# Test-Time Scaling of Diffusion Models via Noise Trajectory Search

**Vignav Ramesh**
Harvard University
vignavramesh@college.harvard.edu

**Morteza Mardani**
NVIDIA
mmardani@nvidia.com

## Abstract

The iterative and stochastic nature of diffusion models enables *test-time scaling*, whereby spending additional compute during denoising generates higher-fidelity samples. Increasing the number of denoising steps is the primary scaling axis, but this yields quickly diminishing returns. Instead, optimizing the *noise trajectory*—the sequence of injected noise vectors—is promising, as the specific noise realizations critically affect sample quality; but this is challenging due to a high-dimensional search space, complex noise-outcome interactions, and costly trajectory evaluations. We address this by first casting diffusion as a Markov Decision Process (MDP) with a terminal reward, showing tree-search methods such as Monte Carlo tree search (MCTS) to be meaningful but impractical. To balance performance and efficiency, we then resort to a relaxation of MDP, where we view denoising as a sequence of independent *contextual bandits*. This allows us to introduce an $\epsilon$-greedy search algorithm that *globally explores* at extreme timesteps and *locally exploits* during the intermediate steps where de-mixing occurs. Experiments on EDM and Stable Diffusion reveal state-of-the-art scores for class-conditioned/text-to-image generation, exceeding baselines by up to $164\%$ and matching/exceeding MCTS performance. To our knowledge, this is the first practical method for test-time noise *trajectory* optimization of *arbitrary (non-differentiable)* rewards.

## 1 Introduction

Diffusion models have emerged as the de-facto standard for generative modeling of visual and, more recently, language data [Ho et al., 2020, Sohl-Dickstein et al., 2015, Nie et al., 2025]. A notable property of these models is their *test-time scaling*: the iterative and stochastic nature of the generation process allows for enhanced sample quality by allocating additional computational resources during inference. A primary strategy for leveraging this has been to increase the number of *denoising steps*. However, empirical observations indicate that performance gains from more denoising steps typically plateau after a few dozen iterations [Karras et al., 2022, Song et al., 2022, 2021].

The diminishing returns from merely increasing denoising steps motivate exploring alternative test-time optimizations. Optimizing the *noise trajectory* itself is a promising direction. Diffusion models iteratively transform an initial noise distribution towards a target data distribution via a stochastic process (often a stochastic differential equation (SDE)), where sequentially introduced fresh noise instances define a random path. The characteristics of these injected noises, forming the *noise trajectory*, are crucial to sample quality.

Recent studies show that injected noises are not equally effective; specific noise realizations can produce higher fidelity samples, with certain noises better suited for different objectives [Qi et al., 2024, Ahn et al., 2024]. However, optimizing this *noise trajectory* is challenging: the search

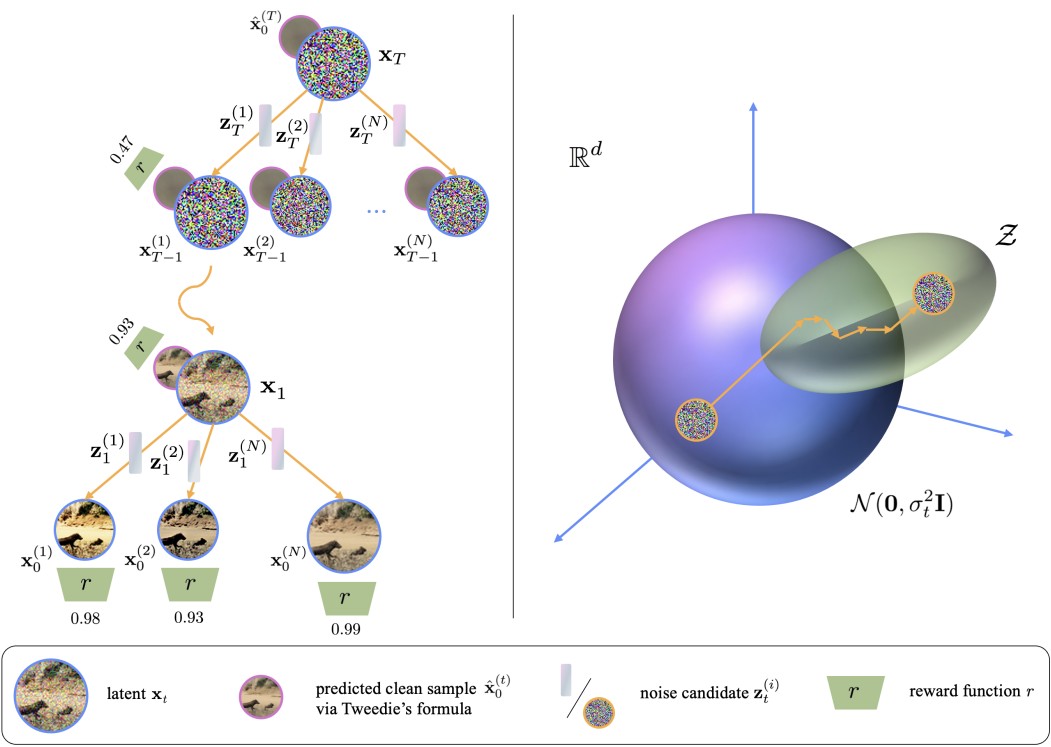

Figure 1: (*Left*) Implicit denoising tree traversed by search algorithms. (*Right*) Visualization of local search in noise space to maximize reward at a single timestep $t$.

space is vast, the noise affects the generation intricately, and evaluating entire trajectories is often computationally prohibitive.

Recent works have explored this noise search; notably, Ma et al. [2025] explores a zero-order *local* search of candidates for the initial noise in ODE-based diffusion sampling. However, *no work as of yet* has explored higher-order search over the *entire* noise trajectory for SDE-based diffusion models.

To develop a framework for finding the optimal trajectory of noise over $T$ denoising steps, we first frame the diffusion sampling process as a $T$-step Markov decision process (MDP) with a terminal reward that is generally non-differentiable. This allows us to explore sophisticated tree search algorithms including a Monte Carlo tree search (MCTS) over the denoising tree; however, MCTS—while a pseudo-upper bound on performance in its approximation of an exhaustive search over the denoising tree—is limited to a pre-selected set of noise candidates per timestep, and is quite computationally expensive (Table 3).

To address this, we introduce a *relaxation* of the aforementioned MDP by formulating each of the $T$ denoising steps as an independent contextual bandit problem. This approximation, which disregards inter-step dependencies inherent in the full MDP, enables more computationally tractable search strategies. Within this bandit framework, we propose a $\epsilon$-greedy variant of zero-order search. This method is designed to strike an effective balance between *global exploration* of the noise space, particularly at extreme (initial and final) timesteps, and more focused *local exploration* during the critical intermediate denoising steps where demixing and mode selection occur predominantly [Karras et al., 2022, Hang et al., 2024, Wang et al., 2025b].

We evaluate our proposed test-time scaling approach on the Elucidated Diffusion Model (EDM) [Karras et al., 2022] for class-conditional image generation (on ImageNet) and Stable Diffusion [Rombach et al., 2022] for text-to-image generation. Our experiments employ task-specific, generally non-differentiable rewards, e.g. a classifier-based reward for EDM to measure class adherence, and CLIP score to assess prompt-image alignment for Stable Diffusion. In both cases, our $\epsilon$-greedy noise trajectory search method achieves state-of-the-art generation quality. It significantly outperforms established baselines, including zero-order search [Ma et al., 2025], rejection sampling [Chatterjee

and Diaconis, 2017], and beam search [Fernandes et al., 2025]. Notably, our computationally efficient approach matches and exceeds MCTS, demonstrating its efficacy in navigating the noise trajectory search space.

All in all, our key contributions are summarized as follows:

- We propose an effective **test-time** noise-trajectory search for diffusion models. Inspired by contextual bandits, it employs an $\epsilon$-greedy search to optimize *arbitrary (non-differentiable) rewards* and *empirically outperforms MCTS*.

- We provide evidence and theoretical intuition that our $\epsilon$-greedy search algorithm **explores globally and exploits locally**. Notably, the search tends to be more local during middle diffusion steps where de-mixing occurs, and more global at large and small noise levels.

- We perform extensive comparisons with alternative search algorithms, including zero-order search, best-of-$N$ sampling, beam search, and MCTS. These comparisons reveal the importance of adapting the search strategy (local vs. global) based on the diffusion step.

- We demonstrate the efficacy of our method for noise optimization on **EDM** and **Stable Diffusion**. Under various rewards (e.g. classifier probability, VLM prompt-image alignment), our approach achieves up to **164%** improvement in sample quality over vanilla generation.

## 2 Related Work

**Reward optimization with diffusion models.** Several works have explored direct fine-tuning of diffusion models with reward functions or human preferences [Black et al., 2024, Domingo-Enrich et al., 2025, Wallace et al., 2023, Wang et al., 2025a]. However, these approaches require expensive training and tie the model to a single reward. Alternatively, Song et al. [2021] and Bansal et al. [2023] have explored reward gradient-based guidance; but this is limited to differentiable reward functions and continuous-state diffusion models. Therefore, steering arbitrary diffusion models at inference time with arbitrary rewards remains a challenge; our work addresses this with the implementation of more sophisticated step-level, gradient-free search algorithms used solely at inference time.

**Test-time scaling.** While test-time scaling has been predominantly explored for language models [Yao et al., 2023, Muennighoff et al., 2025], recent work has begun to investigate test-time scaling behavior for diffusion. One means of investing more compute in diffusion model sampling is increasing the number of denoising steps; this generally leads to better generations, but with diminishing benefits, due to accumulation of approximation and discretization errors [Xu et al., 2023b]. In this work, we fix the denoising steps and explore scaling along an orthogonal axis: compute devoted to searching for *more favorable noise candidates* during the sampling process.

Few past works have considered this search problem. Ma et al. [2025] propose a framework for searching for optimal noise in diffusion model sampling, characterized by the choice of algorithm (e.g. best-of-$N$ search, zero-order search, and "search over paths") as well as reward function (e.g. FID [Heusel et al., 2018], prompt-image embedding similarity [Radford et al., 2021, Caron et al., 2021]). However, all experiments are limited to the ODE formulation of diffusion models, which means that the mapping from the initial noise $\mathbf{x}_T$ to the clean sample $\mathbf{x}_0$ is entirely deterministic; that is, Ma et al. [2025] only have to search over the choices for $\mathbf{x}_T$. In this work, we extend this to the SDE formulation and consider an exponentially larger search space: noise candidates at *every denoising step*, $\{\mathbf{z}_t\}_{t=1}^{T}$. Hence this opens up an investigation of more sophisticated step-level search algorithms (e.g. MCTS), which to our knowledge has *never before been explored*.

Zhang et al. [2025] propose T-SCEND, which sequentially performs best-of-$N$ random search and MCTS as denoising proceeds; however this paper does not truly implement standard MCTS as (1) at timestep $t$ they estimate reward by computing the predicted clean sample $\hat{\mathbf{x}}_0^{(t)}$ via Tweedie's formula [Efron, 2011] (which estimates the posterior mean $\mathbb{E}[\mathbf{x}_0 \mid \mathbf{x}_t, \mathbf{c}]$ from a noisy sample $\mathbf{x}_t$, see Appendix B) rather than doing a complete rollout and then backpropagating reward on the true clean sample, making their method still a greedy search; and (2) their method performs best-of-$N$ search for the majority of denoising steps, only doing their "MCTS" for a small number ($\sim$3-5) of denoising steps near the very end of the sampling process. This version of MCTS can no longer be considered a true approximation of exhaustive tree search.

# 3 Background

## 3.1 Diffusion Models

We consider conditional diffusion models that learn a distribution $p(\mathbf{x}_0 \mid \mathbf{c})$ given data samples $\mathbf{x}_0$ and contexts $\mathbf{c}$. The continuous-time forward process gradually perturbs clean data $\mathbf{x}_0$ towards a noise distribution. This can be described by the stochastic differential equation (SDE):

$$d\mathbf{x}_t = \sqrt{2\sigma(t)\dot{\sigma}(t)}d\boldsymbol{\omega}_t, \tag{1}$$

where $\boldsymbol{\omega}_t$ is a standard Wiener process and $\sigma(t)$ is a monotonically increasing noise schedule with $\sigma(0) = 0$. This process yields $\mathbf{x}_t \sim \mathcal{N}(\mathbf{x}_0, \sigma(t)^2\mathbf{I})$. For a sufficiently large $\sigma(T)$ (e.g. $\sigma(T) = 1$ for normalized data), the final distribution $p(\mathbf{x}_T \mid \mathbf{c}) \approx \mathcal{N}(\mathbf{0}, \sigma(T)^2\mathbf{I})$.

Sampling begins with drawing $\mathbf{x}_T \sim \mathcal{N}(\mathbf{0}, \sigma(T)^2\mathbf{I})$ and then simulating the corresponding reverse time SDE to obtain a clean sample $\mathbf{x}_0$. This reverse process is given by:

$$d\mathbf{x}_t = -2\dot{\sigma}(t)\sigma(t)\nabla_{\mathbf{x}_t} \log p(\mathbf{x}_t \mid \sigma(t), \mathbf{c})dt + \sqrt{2\dot{\sigma}(t)\sigma(t)}d\bar{\boldsymbol{\omega}}_t, \tag{2}$$

where $d\bar{\boldsymbol{\omega}}_t$ is a reverse-time Wiener process and $\dot{\sigma}(t) = d\sigma/dt$. The score function $\nabla_{\mathbf{x}_t} \log p(\mathbf{x}_t \mid \sigma(t), \mathbf{c})$ is approximated by a neural network $D_\theta(\mathbf{x}_t, \sigma(t), \mathbf{c})$. The choice of sampler, which discretizes Eq. (2), is also crucial. In our experiments, we use either the EDM sampler (Alg. 2) [Karras et al., 2022] for class-conditional generation or the DDIM sampler (Alg. 3) [Song et al., 2022] for text-to-image generation, enabling high-quality synthesis with few steps $T$ (e.g. 18 for EDM, 50 for DDIM).

## 3.2 From Markov Decision Processes to Contextual Bandits

A Markov Decision Process (MDP) formalizes sequential decision-making problems. An MDP is defined by a tuple $(\mathcal{S}, \mathcal{A}, \rho_0, P, R)$, where $\mathcal{S}$ is the state space, $\mathcal{A}$ is the action space, $\rho_0$ is the initial state distribution, $P(\mathbf{s}_{t+1} \mid \mathbf{s}_t, \mathbf{a}_t)$ is the state transition probability kernel, and $R(\mathbf{s}_t, \mathbf{a}_t)$ is the reward function. At each timestep $t$, an agent observes a state $\mathbf{s}_t \in \mathcal{S}$, takes an action $\mathbf{a}_t \in \mathcal{A}$ according to its policy $\pi(\mathbf{a}_t \mid \mathbf{s}_t)$, receives a reward $r_t = R(\mathbf{s}_t, \mathbf{a}_t)$, and transitions to a new state $\mathbf{s}_{t+1} \sim P(\cdot \mid \mathbf{s}_t, \mathbf{a}_t)$.

A continuous contextual bandit problem can be viewed as a specialized single-state, one-step MDP with a continuous action space. It is defined by a tuple $(\Phi, \mathcal{A}, \tilde{R})$, where $\Phi$ is the space of contexts, $\mathcal{A}$ is the continuous action space (shared with the definition of MDP), and $\tilde{R}$ is the reward function specific to the bandit setting. In each round $i$, the agent observes a context $\phi_i \in \Phi$ (analogous to an initial state $\mathbf{s}_0$ from $\rho_0$ in an MDP, but where contexts for different rounds are typically drawn independently). The agent then selects an action $\mathbf{a}_i \in \mathcal{A}$ based on its policy $\pi(\mathbf{a}_i \mid \phi_i)$ and immediately receives a reward $\tilde{r}_i = \tilde{R}(\phi_i, \mathbf{a}_i)$.

# 4 Searching Over Noise Trajectories

Standard diffusion samplers use random noise per reverse step. To optimize generation for a downstream reward $r(\mathbf{x}_0, \mathbf{c})$, we instead search for the optimal $T$-step noise trajectory $\mathbf{Z} := \begin{bmatrix} \mathbf{z}_T & \mathbf{z}_{T-1} & \cdots & \mathbf{z}_1 \end{bmatrix}^\top$. Given an initial $\mathbf{x}_T \sim \mathcal{N}(\mathbf{0}, \sigma(T)^2\mathbf{I})$, a pre-trained model $D_\theta$, context $\mathbf{c}$, and a chosen sampler, the map from $\mathbf{Z}$ to the final sample $\mathbf{x}_0(\mathbf{Z}, \mathbf{c}, \mathbf{x}_T)$ is deterministic.

The search space $\mathbf{Z} \in \mathbb{R}^{T \times d}$ (where $d$ is the noise dimension, i.e. $CHW$ where $C \times H \times W$ is the image resolution) is high-dimensional and unbounded; optimization typically explores specific regions or targeted deviations from standard Gaussian noise (Figure 1).

## 4.1 Diffusion as a Finite-Horizon MDP

To formalize this search, we cast the $T$-step reverse diffusion process (denoising steps indexed $t = T, \ldots, 1$) as a Markov Decision Process (MDP) with a terminal reward:

- **States:** $\mathbf{s}_t = (\mathbf{c}, \mathbf{x}_t, t)$, with $t \in \{T, \ldots, 0\}$ indexing the reverse diffusion step.

- **Actions:** $\mathbf{a}_t = \mathbf{z}_t$, the noise vector for $t \in \{T, \ldots, 1\}$.
- **Initial State:** $\mathbf{s}_T = (\mathbf{c}, \mathbf{x}_T, T)$, where $\mathbf{c} \sim p(\mathbf{c})$ and $\mathbf{x}_T \sim \mathcal{N}(\mathbf{0}, \sigma(T)^2\mathbf{I})$.
- **Transitions:** Deterministic. State $\mathbf{s}_t$ and action $\mathbf{a}_t = \mathbf{z}_t$ yield next state $\mathbf{s}_{t-1} = (\mathbf{c}, f(\mathbf{x}_t, t, \mathbf{c}, \mathbf{z}_t), t - 1)$. The function $f$ is one step of the chosen sampler (e.g. lines 2-6 of Alg. 2, lines 2-3 of Alg. 3) using $\mathbf{z}_t$ as the injected noise.
- **Reward:** Terminal reward $R = r(\mathbf{x}_0, \mathbf{c})$ at state $\mathbf{s}_0$ (after action $\mathbf{a}_1$); zero otherwise. $r(\cdot, \cdot)$ evaluates the generated sample $\mathbf{x}_0$, potentially with respect to the conditioning $\mathbf{c}$.

Finding the optimal $\mathbf{Z}$ can be written as solving:

$$\max_{\mathbf{Z} \in \mathbb{R}^{T \times d}} r(\mathbf{x}_0(\mathbf{Z}, \mathbf{c}, \mathbf{x}_T), \mathbf{c}), \tag{3}$$

where $\mathbf{x}_0(\mathbf{Z}, \mathbf{c}, \mathbf{x}_T)$ is the final sample generated using the noise sequence $\mathbf{Z}$.

## 4.2 Monte-Carlo Tree Search (MCTS)

In this general formulation, the space of noise candidates $\mathbf{Z} \in \mathbb{R}^{T \times d}$ is extremely large; hence, we first consider the simplification where we predecide a fixed set of $N$ noise candidates $\{\mathbf{z}_t^{(i)}\}_{i=1}^N$ for each timestep $t$. This gives rise to a "denoising tree" with depth $T$ and branching factor $N$ as shown in Figure 1, where each node $\mathbf{x}_t$ is a latent in the sampling process and each edge is a noise candidate $\mathbf{z}_t^{(i)}$. Finding the best noise trajectory now corresponds to finding the best root-to-leaf path in this tree; hence, we can directly apply tree-search algorithms to this setting, including Monte Carlo tree search (MCTS) as detailed in Algorithm 4.

MCTS approximates exhaustive search over the $N^T$ possible noise trajectories, with the approximation error going to 0 as the number of simulations $S \to \infty$. Full MCTS though faces two key drawbacks: (1) it is extremely computationally expensive, with its number of function evaluations (NFEs)—i.e. number of forward passes through $D_\theta$—scaling *quadratically* in the number of denoising steps $T$ (see Table 3 for each sampling method's NFE formula); and (2) MCTS by definition is restricted to a fixed denoising tree, i.e. at each timestep $t$ we cannot explore "noise space" beyond the $N$ initially determined candidates. This is a critical issue, as the initial $N$ candidates are sampled uniformly from our prior (i.e., $\mathcal{N}(\mathbf{0}, \sigma_t^2\mathbf{I})$)) as opposed to any higher-reward regions of noise space that might exist. In other words, we have no notion of importance sampling; we are allocating the entire computation cost of running MCTS to samples that in general, likely yield lower-than-optimal rewards [Chatterjee and Diaconis, 2017].

To address these challenges, next we posit an alternative formulation of the denoising process that *relaxes* the previous MDP.

## 4.3 Noise Search via Contextual Bandits

Recall that the SDE diffusion formulation requires the injected noise samples at each timestep to be independent (and Normally distributed). Thus, if we assume independence of actions across timesteps, and ignore the impact of the action at timestep $t$ on the state at $t - 1$, we can reformulate denoising as a sequence of $T$ independent continuous contextual bandit problems; at each timestep $t$, we have the bandit characterized by:

$$\phi_i \triangleq (\mathbf{c}, \mathbf{x}_t, t) \qquad \mathbf{a}_i \triangleq \mathbf{z}_t \qquad \tilde{R}(\phi_i, \mathbf{a}_i) \triangleq r\left(\hat{\mathbf{x}}_0^{(t)}\right)$$

where $\hat{\mathbf{x}}_0^{(t)} = \mathbb{E}[\mathbf{x}_0 \mid \mathbf{x}_t, \mathbf{z}_t, \mathbf{c}]$ is computed via Tweedie's formula.

The "sequence-of-bandits" formulation for denoising introduces a key tradeoff. By ignoring interactions across steps (i.e., how $\mathbf{z}_t$ affects $\mathbf{x}_{t-1}$ which in turn informs the choice of $\mathbf{z}_{t-1}$), and computing reward on our time-$t$ predicted clean samples $\hat{\mathbf{x}}_0^{(t)}$ rather than our actual clean samples $\mathbf{x}_0$, we have a worse approximation of our original optimization problem; but we in turn have vastly more NFEs to spend on exploration at each timestep (i.e., those that were allocated to simulation/backpropagation in MCTS), and in particular can search over more of noise space than the fixed $N$ candidates we were previously limited to.

---

**Algorithm 1** $\epsilon$-greedy noise search

---

**Require:** Discretization timesteps $T$, context (e.g. class label) $\mathbf{c}$, learned denoising network $D_\theta$, max. step size scaling factor $\lambda$, number of local search iterations $K$, branching factor (number of noise candidates) $N$, sampling step function $f$, mixture proportion $\epsilon$, reward function $r$, initial noise sample $\mathbf{x}_T$

1: **for** $t = T$ to 1 **do**

   **Sample** $\mathbf{p} \sim \mathcal{N}(\mathbf{0}, \sigma_t^2 \mathbf{I})$ // pivot

2:   **for** $k = 1$ to $K$ **do**

   $\forall\, i = 1, \ldots, N$ sample noise candidate $\mathbf{z}_t^{(i)}$ as follows: with probability $\epsilon$ sample $\mathbf{z}_t^{(i)} \sim \mathcal{N}(\mathbf{0}, \sigma_t^2 \mathbf{I})$, else let $\mathbf{z}_t^{(i)} = \mathbf{u}\sqrt{2d}\mathbf{z} + \mathbf{p}$ where $\mathbf{z} \sim \mathcal{N}(\mathbf{0}, \mathbf{I})$, $\mathbf{u} \sim \mathrm{Unif}(0, \lambda)$

   Set $\mathbf{p} = \mathbf{z}_t^{(i)}$ for $i$ s.t. one-step $\hat{\mathbf{x}}_0$ prediction using $\mathbf{z}_t^{(i)}$ attains highest score under $r$

3:   **end for**

   Set $\mathbf{x}_{t-1} = f(\mathbf{x}_t, t, \mathbf{c}, \mathbf{p})$

4: **end for**

5: **return** $\mathbf{x}_0$

---

### 4.4 $\epsilon$-greedy

Formally, with sequence-of-bandits denoising, we can explore "greedy algorithms" that attempt to independently solve at each timestep $t$:

$$\max_{\mathbf{z}_t \in \mathbb{R}^d} r(\hat{\mathbf{x}}_0^{(t)}, \mathbf{c}), \qquad t = 1, \ldots, T. \tag{4}$$

Inspired by the semi-uniform strategies originally proposed to approximately solve bandit problems [Slivkins, 2024], including the original $\epsilon$-greedy bandit algorithm [Sutton and Barto, 1998], we introduce a novel sampling algorithm that is an $\epsilon$-contaminated version of zero-order search (henceforth simply "$\epsilon$-greedy" (Alg. 1), where at each timestep $t$: (1) We start with a random Gaussian noise $\mathbf{p} \sim \mathcal{N}(\mathbf{0}, \sigma_t^2 \mathbf{I})$ as our *pivot*. (2) We then find $N$ candidates $\{\mathbf{z}_t^{(i)}\}_{i=1}^N$, via the following procedure: with probability $\epsilon$, $\mathbf{z}_t^{(i)} \sim \mathcal{N}(\mathbf{0}, \sigma_t^2 \mathbf{I})$; else, we draw a random $\mathbf{z}_t^{(i)}$ from the pivot's neighborhood ($\lambda\sqrt{2d}$-radius ball around $\mathbf{p}$). (3) We compute the clean sample predicted $\hat{\mathbf{x}}_0^{(i)}$ using Tweedie's formula using each candidate $\mathbf{z}_t^{(i)}$. (4) Finally, we score each clean sample, set the pivot to the noise candidate leading to the clean sample with the highest scoring, and repeat steps 1-3 $K-1$ times, finally setting $\mathbf{z}_t$ to the last value of $\mathbf{p}$.

This simple change from the neighborhood in zero-order search to the $\epsilon$-contaminated mixture distribution

$$\epsilon\mathcal{N}(\mathbf{0}, \sigma_t^2 \mathbf{I}) + (1 - \epsilon)\mathcal{N}(\mathbf{p}, 2\lambda^2 d\mathbf{I})$$

yields remarkable performance improvements, which we discuss in section 5. We also provide a theoretical formalization of our algorithm's effectiveness in Appendix D.

## 5 Experiments

### 5.1 Class-Conditional Image Generation

To further elucidate our design choices, we start by presenting a *design walk-through* for the **class-conditional ImageNet generation** task. We adopt EDM [Karras et al., 2022] pre-trained on ImageNet with resolution $64 \times 64$ and perform sampling with the stochastic EDM sampler (Alg. 2). The number of denoising steps is fixed to 18. Unless otherwise specified, we use the classifier-free guidance 1.0, focusing on the simple conditional generation task without guidance [Ho and Salimans, 2022] .

#### 5.1.1 Baselines

As baselines, we consider the following algorithms (see details in Appendix C):

**Rejection (best-of-$N$) sampling [Chatterjee and Diaconis, 2017].** We randomly sample $N$ root-to-leaf paths of the *denoising tree*, and choose the path that produces $\mathbf{x}_0$ with the highest reward.

Table 1: **EDM results by sampling method.** Each value in columns 2-4 is obtained by generating 36 images given random ImageNet class labels. The column header denotes the reward used both to score the final images and during sampling if applicable. We set $\lambda = 0.15$ and $\epsilon = 0.4$. Note that we measure distance as $\|\mathbf{z}_t^{(i)} - \mathbf{p}\|_F$, hence let $\lambda$ a scaling factor applied to $\mathbb{E}_{\mathbf{x},\mathbf{y}\sim\mathcal{N}(0,1)^d}[\|\mathbf{x} - \mathbf{y}\|_F] \approx \sqrt{2d}$ (where $\|\cdot\|_F$ is the Frobenius norm. Re. notation, $N$ is branching factor, $B$ is beam width, $S$ is number of MCTS simulations, and $K$ is the number of local search iterations. Generating each sample took $<1$ second for naive sampling (the lowest-compute method) and $<1$ minute for MCTS (the highest-compute method) on a single A100 (40GB). We provide error bars for each reward and sampling method, computed as the standard deviation of scores for a given prompt over 20 generations with variability from randomness of the noise draws.

| Method | Brightness | Compressibility | Classifier | NFEs |
|---|---|---|---|---|
| Naive Sampling | $0.4965\pm0.01$ | $0.3563\pm0.07$ | $0.3778\pm0.04$ | 18 |
| Best of 4 Sampling | $0.5767\pm0.01$ | $0.4220\pm0.02$ | $0.5461\pm0.00$ | 72 |
| Beam Search ($N = 4, B = 2$) | $0.6334\pm0.02$ | $0.4679\pm0.05$ | $0.5536\pm0.02$ | 144 |
| MCTS ($N = 4, S = 8$) | $0.7575\pm0.02$ | $0.5395\pm0.04$ | $0.9666\pm0.03$ | 3888 |
| Zero-Order Search ($N = 4, K = 20$) | $0.6083\pm0.01$ | $0.3751\pm0.02$ | $0.6261\pm0.04$ | 1440 |
| **$\epsilon$-greedy** ($N = 4, K = 20$) | $\mathbf{0.9813}\pm0.01$ | $\mathbf{0.7208}\pm0.03$ | $\mathbf{0.9885}\pm0.04$ | 1440 |

**Beam search [Fernandes et al., 2025].** We start at the root and at each node $\mathbf{x}_t$, traverse the edge to the child $\mathbf{x}_{t-1}^{(j)}$ whose corresponding $\hat{\mathbf{x}}_0^{(t-1)}$ (obtained via Tweedie's formula [Efron, 2011]) has the highest reward.

**Zero-order search [Ma et al., 2025].** At each timestep, we start with a random pivot and hill-climb with step size scaling factor $\lambda$ for $K$ iterations, greedily selecting the best candidate at each iteration. This can be seen as $\epsilon$-greedy with $\epsilon = 0$, and thus no global exploration.

### 5.1.2 Rewards

We consider the following reward functions:

**Brightness.** We compute the $[0, 1]$-normalized luminance formula for perceived brightness, defined for an RGB image as $0.2126R + 0.7152G + 0.0722B$.

**Compressibility.** Following Black et al. [2024], we calculate the number of bytes $b$ of the maximally-compressed JPEG encoding of the generated images and then bound this in $[0, 1]$-range via the formula $1 - \frac{b_{\max}-b}{b}$, where $b_{\max} = 3000$ is manually set based on empirical observations of 50 compressed real ImageNet images. Note that since we fix the resolution of diffusion model samples at $64 \times 64$, the file size is determined solely by the compressibility of the image.

**Class probability under ImageNet classifier.** We adopt the ImageNet-1k $64 \times 64$ classifier of Dhariwal and Nichol [2021] for classifier-guided diffusion. For each generated image–label pair, the reward is the classifier's probability assigned to the ground-truth class. The classifier is the downsampling trunk of a U-Net, with an attention pool at the $8 \times 8$ layer producing the final logits.

### 5.1.3 Results

Table 1 displays class-conditional image generation results. We discuss key observations below:

**MCTS as an approximate upper bound.** Recall that rejection sampling, beam search, and MCTS are all methods to choose a *root-to-leaf* path in a fixed denoising tree. Beam search, by virtue of greedily selecting the best noise candidate at each timestep independently, fails to consider interactions across trajectories; and rejection sampling, while considering trajectories as a whole, explores a very small number of them. (Indeed, rejection sampling faces the same issue as MCTS whereby we have no notion of importance sampling—we are allocating all computation to $N$ trajectories that, in general, likely yield lower-than-optimal rewards [Chatterjee and Diaconis, 2017]; but this issue is far more pronounced for rejection sampling due to the limited number of trajectories seen during sampling).

MCTS solves for these drawbacks by approximating exhaustive search over the $N^T$ possible root-to-leaf paths in the tree, with the approximation error going to 0 as the number of simulations $S \rightarrow \infty$.

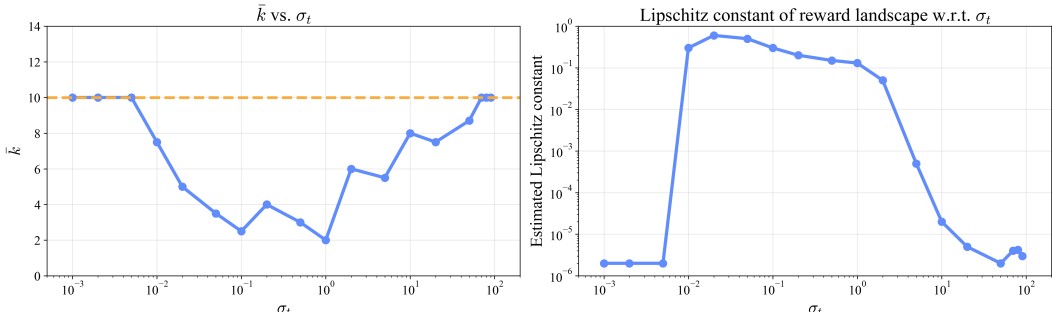

Figure 2: *(Left)* Average local-search iteration at which a random Normal candidate is chosen $(\bar{k})$ as a function of $\sigma_t$. *(Right)* Estimated Lipschitz constant of the ImageNet reward as a function of $\mathbf{x}_t$, across $\sigma_t$ values.

We indeed see this in Table 1: rejection sampling and beam search beat naive sampling across all reward functions; but excluding zero-order search, *MCTS is best across the board, as expected*.

**Leaving the denoising tree.** Importantly, recall that rejection sampling, beam search, and MCTS all optimize a *fixed* denoising tree, i.e. at each timestep $t$ we cannot explore "noise space" beyond the $N$ initially determined candidates. Zero-order search and $\epsilon$-greedy address this, with the amount of exploration in noise space dictated by the number of local search iterations $K$, the maximum step size scaling factor $\lambda$, and the *global exploration factor* $\epsilon$. Table 1 displays results for both these methods; somewhat surprisingly, the vanilla zero-order search lags behind the MCTS performance, while $\epsilon$-greedy far exceeds it.

**Explore globally and exploit locally.** The above results lead us to ask: *why does simply drawing a Normal sample w.p. $\epsilon$ while otherwise sampling from the pivot's neighborhood (w.p. $1 - \epsilon$) yield such remarkable improvement*?

We hypothesize the following: at time $t$, conditioned on latent $\mathbf{x}_t$, there is a specific "region" in the noise space (call it $\mathcal{Z}$) containing noise realizations that best fit the downstream objective (e.g. the green region in Figure 1). In particular, $\mathcal{Z}$ is not necessarily a subset of $\mathcal{N}(\mathbf{0}, \sigma_t^2 \mathbf{I})$, although it can.

During vanilla zero-order search, at each local search iteration of each timestep we are restricted to the $\lambda\sqrt{2d}$-radius ball around the original pivot, which is simply a random draw from $\mathcal{N}(\mathbf{0}, \sigma_t^2 \mathbf{I})$. More likely than not, the pivot $\mathbf{p} \notin \mathcal{Z}$; and over $K$ local search iterations, we move an expected distance of $K\mathbb{E}_{\mathbf{u}\sim\text{Unif}(0,\lambda)}[\mathbf{u}]\sqrt{2d} = K\lambda\sqrt{d/2}$ away from $\mathbf{p}$. But with $\epsilon$-greedy, we can explore the entire Normal with $\epsilon$ probability; a few such "random jumps" can move us into $\mathcal{Z}$, at which point we see large returns from hill-climbing. Indeed, we see this empirically: for each local search iteration $k \in \{1, \dots, K\}$ of $\epsilon$-greedy, let $I_k = \mathbb{I}(\epsilon\text{-greedy selects one of the random Normal draws instead of all the neighborhood draws in iteration } k)$. Define $\bar{k} = \sum_{k=1}^{K}(k-1)I_k$. Note that if $\epsilon$-greedy chooses one of the random draws at every iteration, $\bar{k} = 10$; but if $\bar{k} < 10$, $\epsilon$-greedy only chooses the random draw in early iterations $k$ and only hill-climbs (chooses candidates from the neighborhood) at later iterations. Figure 2 (left) plots $\bar{k}$ as a function of $\sigma_t$. For initial denoising steps, the random noise is always chosen, hence $\bar{k} = 10$. But in the intermediate denoising steps, $\bar{k} \ll 10$—supporting our hypothesis that we initially need a couple of random draws to get to $\mathcal{Z}$, but for all subsequent iterations, *hill climbing is required for us to potentially leave the Normal distribution and find noises that maximize the reward*.

The aforementioned findings raise the question: in $\epsilon$-greedy, why do we only benefit from hill-climbing at *intermediate denoising steps*? We posit that the reward landscape as a function of $\mathbf{x}_t$ is much smoother, perhaps near flat, at extreme time steps rather than intermediate ones, hence there is limited performance gain from hill climbing at those steps. This hypothesis aligns with the well-known result that denoising is harder at intermediate timesteps as this is where demixing occurs (the result is already well-refined at later denoising steps, and latents are too noisy to meaningfully understand the features of the image at earlier timesteps) [Karras et al., 2022, Hang et al., 2024, Wang et al., 2025b]. To test this, we measure the *sensitivity* of the reward landscape as a function of $\mathbf{x}_t$ for each $t$; specifically, we proxy the sensitivity with the Lipschitz constant estimator $\|\nabla_{\mathbf{x}_t} r(\hat{\mathbf{x}}_0^{(t)}, \mathbf{c})\|_2$,

Table 2: **Stable Diffusion results by sampling method.** Following Ma et al. [2025], we use natural language versions of the same class labels used in EDM generation (e.g. "a toucan"). Generating each sample took ~1 second for naive sampling (the lowest-compute method) and <2 minutes for MCTS (the highest-compute method) on a single A100 (40GB). We again provide as error bars the standard deviation of scores for a given prompt over 20 generations with variability due to randomness of the noise draws.

| Method | Brightness | Compressibility | CLIP | NFEs |
|---|---|---|---|---|
| Naive Sampling | $0.4800{\pm}0.02$ | $0.6314{\pm}0.03$ | $0.2457{\pm}0.00$ | 18 |
| Best of 4 Sampling | $0.5160{\pm}0.04$ | $0.6847{\pm}0.02$ | $0.2673{\pm}0.00$ | 72 |
| Beam Search ($N = 4, B = 2$) | $0.4981{\pm}0.01$ | $0.7014{\pm}0.01$ | $0.2682{\pm}0.00$ | 144 |
| MCTS ($N = 4, S = 8$) | $0.6077{\pm}0.03$ | $0.7322{\pm}0.02$ | $0.2885{\pm}0.01$ | 3888 |
| Zero-Order Search ($N = 4, K = 20$) | $0.5764{\pm}0.02$ | $0.7281{\pm}0.06$ | $0.2689{\pm}0.02$ | 1440 |
| **$\epsilon$-greedy** ($N = 4, K = 20$) | **$0.7110$**${\pm}0.03$ | **$0.8052$**${\pm}0.05$ | **$0.2973$**${\pm}0.02$ | 1440 |

i.e. the local gradient norm of the reward function w.r.t. $\mathbf{x}_t$. Figure 2 (right) shows a plot of the estimated Lipschitz constant by $\sigma_t$, in which we indeed see that the reward is most sensitive to $\mathbf{x}_t$ perturbations at intermediate steps. *To our knowledge, this is the first work to demonstrate the importance of adapting the search strategy – local vs. global – based on the diffusion timestep.*

Additional experiments, as well as qualitative results, are included in Appendices F, G and H. Notably, we conduct hyperparameter sweeps and measure how each sampling method scales with increasing NFEs in Appendix G, finding that *$\epsilon$-greedy achieves the best scaling laws empirically amongst all sampling methods.*

## 5.2 Extending to Text-to-Image Generation

Next, we extend our experiments to the more difficult task of **text-to-image generation**. We use Stable Diffusion v1.5 [Rombach et al., 2022], setting all algorithm hyperparameters to their optimal values from EDM. We enable classifier-free guidance (default scale 7.5) and set $T = 50$ here as is typical with DDIM sampling. The resolution of the generated images is $512 \times 512$.

In lieu of the ImageNet classifier reward, we consider a very general purpose reward function for training a text-to-image model: *prompt-image alignment*. However, specifying a reward that captures generic prompt alignment is difficult, conventionally requiring large-scale human labeling efforts. Following Ma et al. [2025], we proxy human ratings of prompt-image alignment with a direct measure of such alignment in joint image-text embedding space; namely, we compute the cosine similarity between the CLIP representations of both the image and original prompt [Radford et al., 2021]. Samples that more faithfully adhere to the semantics of the prompt should receive higher rewards under this framework, to the extent that those visual details are represented distinctly in CLIP embedding space.

The Stable diffusion results (Table 2) indeed affirm the superiority of our $\epsilon$-greedy search, which consistently outperforms other sampling methods across all reward functions. Specifically, $\epsilon$-greedy attains improvements of **48/26/20**% over vanilla sampling for the brightness/compressibility/CLIP reward functions, even exceeding MCTS by up to **18**%. This underscores the robustness and generality of our method across diffusion model *architectures* (image vs. latent models), number of *denoising steps* (18-50), and and context modalities (class label vs. text).

## 6 Conclusion

We present the first practical framework for *test-time noise trajectory optimization* in SDE-based diffusion models. We first cast denoising as an MDP, enabling us to apply sophisticated step-level search algorithms including MCTS; then, via relaxing the MDP to a sequence of independent contextual bandits, we design a novel $\epsilon$-greedy search algorithm that adapts the search strategy (local exploitation vs. global exploration) based on the diffusion step, significantly outperforming vanilla sampling and even MCTS. We extensively discuss broader impacts, limitations, and future work in Appendices I and J.

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

# Contents

# A  Samplers

In all experiments we adopt either the EDM sampler (Alg. 2) [Karras et al., 2022] (for class-conditional generation) or DDIM sampler (Alg. 3) [Song et al., 2022] (for text-to-image generation) as it enables high-quality generation with low number of denoising steps $T$. Note that the choice of sampler fundamentally affects the formulation of denoising as an MDP/sequence-of-bandits problem, and hence our search over noise trajectories, since we define the transition function $P(\mathbf{s}_{t-1} \mid \mathbf{s}_t, \mathbf{a}_t) \triangleq \delta_{(\mathbf{c}, f(\mathbf{x}_t, t, \mathbf{c}, \mathbf{z}_t), t-1)}$ where $\delta_y$ is the Dirac delta function (places all probability mass on single point $y$) and $f$ is one step of the chosen sampler (e.g. lines 2-6 of Alg. 2, lines 2-3 of Alg. 3) using $\mathbf{z}_t$ as the injected noise.

---

**Algorithm 2** EDM Sampling Algorithm

---

**Require:** Number of timesteps $T$, noise levels $\{t_i\}_{i=0}^N$ with $t_0 = t_{\min}$ and $t_N = t_{\max}$, expansion coefficients $\{\gamma_i\}_{i=0}^{N-1}$, initial sample $\mathbf{x}_T \sim \mathcal{N}(\mathbf{0}, t_N^2 \mathbf{I})$, denoising network $D_\theta(\mathbf{x}, t, \mathbf{c})$, $\sigma_t$, noise samples $\{\mathbf{z}_i\}_{i=1}^N \sim \mathcal{N}(\mathbf{0}, \sigma_t^2 \mathbf{I})$

1: **for** $i = T$ **to** 1 **do**
2: $\quad \hat{t}_i = (1 + \gamma_i) t_i; \hat{\mathbf{x}}_i = \mathbf{x}_i + \sqrt{\hat{t}_i^2 - t_i^2}\, \mathbf{z}_i$
3: $\quad \mathbf{d}_i = (\hat{\mathbf{x}}_i - D_\theta(\hat{\mathbf{x}}_i, \hat{t}_i, \mathbf{c})) / \hat{t}_i; \mathbf{x}_{i-1} = \hat{\mathbf{x}}_i + (t_{i-1} - \hat{t}_i)\mathbf{d}_i$
4: $\quad$ **if** $t_{i+1} \neq 0$ **then**
5: $\quad\quad \mathbf{d}_i' = (\mathbf{x}_{i-1} - D_\theta(\mathbf{x}_{i-1}, t_{i-1})) / t_{i-1}; \mathbf{x}_{i-1} = \hat{\mathbf{x}}_i + (t_{i-1} - \hat{t}_i)\left(\frac{1}{2}\mathbf{d}_i + \frac{1}{2}\mathbf{d}_i'\right)$
6: $\quad$ **end if**
7: **end for**
8: **return** $\mathbf{x}_0$

---

**Algorithm 3** DDIM Sampling Algorithm

---

**Require:** Discretization timesteps $T$, diffusion coefficients $\alpha_1, \ldots, \alpha_T$, initial noise $\mathbf{x}_T \sim \mathcal{N}(\mathbf{0}, \mathbf{I})$, context (e.g. text prompt) $\mathbf{c}$, noise vectors $\mathbf{z}_1, \ldots, \mathbf{z}_T \sim \mathcal{N}(\mathbf{0}, \mathbf{I})$, coefficient $\eta \in [0, 1]$ for balancing ODE and SDE, denoising network $D_\theta(\mathbf{x}, t, \mathbf{c})$

1: **for** $t = T$ **to** 1 **do**
2: $\quad \sigma_t = \eta\sqrt{(1 - \alpha_{t-1})/(1 - \alpha_t)}\sqrt{1 - \alpha_t/\alpha_{t-1}}$
3: $\quad \mathbf{x}_{t-1} = \sqrt{\alpha_{t-1}/\alpha_t} \cdot \mathbf{x}_t - \left(\sqrt{\alpha_{t-1}(1 - \alpha_t)/\alpha_t} - \sqrt{1 - \alpha_{t-1} - \sigma_t^2}\right)\epsilon_\theta(\mathbf{x}_t, t, \mathbf{c}) + \sigma_t \mathbf{z}_t$
4: **end for**
5: **return** $\mathbf{x}_0$

---

# B  Stepwise Noise Evaluation via Tweedie's Formula

The greedy algorithms we treat as baselines (beam search, zero-order search), as well as our $\epsilon$-greedy algorithm itself, require evaluating multiple noise candidates $\{\mathbf{z}_t^{(i)}\}_{i=1}^N$ at timestep $t$, independent of future denoising steps $t - 1, \ldots, 1$. We operationalize this evaluation by letting the best noise candidate $\mathbf{z}_t^{(i^*)}$ be that which maximizes the reward of the expected clean sample conditioned on that noise candidate, i.e.,

$$i^* = \arg\max_{i=1,\ldots,N} r(\mathbb{E}[\mathbf{x}_0 \mid f(\mathbf{x}_t, t, \mathbf{c}, \mathbf{z}_t^{(i)})], \mathbf{c}]). \tag{5}$$

We leverage Tweedie's formula [Efron, 2011] to compute the expected clean sample $\hat{\mathbf{x}}_0^{(t)}$ as an estimate of the posterior mean $\mathbb{E}[\mathbf{x}_0 \mid \mathbf{x}_{t-1}, \mathbf{c}]$ from the noisy sample $\mathbf{x}_{t-1}$ produced by doing a denoising step from $\mathbf{x}_t$ with noise candidate $\mathbf{z}_t^{(i)}$. Formally, given $\mathbf{x}_{t-1}, \sigma_{t-1}$, we have

$$\mathbb{E}[\mathbf{x}_0 \mid \mathbf{x}_{t-1}, \mathbf{c}] = \mathbf{x}_{t-1} + \sigma_{t-1}^2 \nabla_{\mathbf{x}_{t-1}} \log p(\mathbf{x}_{t-1} \mid \sigma_{t-1}, \mathbf{c}),$$

where the denoiser $D_\theta$ is the diffusion model (e.g., EDM or Stable Diffusion v1.5) trained to approximate the score function $\nabla_{\mathbf{x}_{t-1}} \log p(\mathbf{x}_{t-1} \mid \sigma_{t-1}, \mathbf{c})$.

## C Search Algorithms

Below, we provide detailed algorithms for the rejection sampling, beam search, zero-order search, and MCTS methods described in the main body of the paper. Table 3 displays the NFEs required per sample generation using each method; notably, all sampling methods but MCTS—including $\epsilon$-greedy, which attains the highest performance across reward functions—require NFEs only *linear* in the number of denoising steps $T$.

---

**Algorithm 4** MCTS noise search

---

**Require:** Discretization timesteps $T$, context (e.g. class label) $\mathbf{c}$, learned denoising network $D_\theta$, branching factor $N$, sampling step function $f$, $[0,1]$-bounded reward function $r$, fixed per-step noise candidates $\{\mathbf{z}_t^{(j)}\}_{(t,j)\in\{1,\ldots,T\}\times\{1,\ldots,N\}}$, number of simulations $S$, exploration constant $C = 1.414$, initial noise sample $\mathbf{x}_T$

$$\text{Define UCB score}(\mathbf{x}_t, t, \mathbf{z}_t^{(j)}) = \frac{\text{reward from } (f(\mathbf{x}_t,t,\mathbf{c},\mathbf{z}_t^{(j)}),t-1)}{\text{\# visits to } (f(\mathbf{x}_t,t,\mathbf{c},\mathbf{z}_t^{(j)}),t-1)} + C\sqrt{\frac{\log(\text{\# visits to } (\mathbf{x}_t,t))}{\text{\# visits to } (f(\mathbf{x}_t,t,\mathbf{c},\mathbf{z}_t^{(j)}),t-1)}}$$

$$\mathcal{C}(\mathbf{x}_t) \triangleq \text{set of child nodes of } \mathbf{x}_t, \text{ s.t. } |\mathcal{C}(\mathbf{x}_t)| \in \{0, N\}, c_j(\mathbf{x}_t) \triangleq f(\mathbf{x}_t, t, \mathbf{c}, \mathbf{z}_t^{(j)})$$

1: **for** $t = T$ to $1$ **do**
2:      **for** $s = 1$ to $S$ **do**
         Let $t' = t$
3:          **while** $|\mathcal{C}(\mathbf{x}_{t'})| \neq 0$ **do**             // selection
$$\text{Let } \mathbf{x}_{t'-1} = f\left(\mathbf{x}_t, t, \mathbf{c}, \mathbf{z}_t^{(\arg\max_j \text{ UCB score}(\mathbf{x}_{t'}, t', \mathbf{z}_{t'}^{(j)}))}\right)$$
         $t' \leftarrow t' - 1$
4:          **end while**
5:          **if** $t' \neq 0$ **then**             // expansion
         $\mathcal{C}(\mathbf{x}_{t'}) \leftarrow \{c_j(\mathbf{x}_{t'})\}_{j=1}^{N}$
6:          **end if**
         For random $k \in \{1, \ldots, N\}$, denoise from $c_k(\mathbf{x}_{t'})$ to get $\hat{\mathbf{x}}_0$             // simulation
7:          **for** $t'' = t' - 1$ to $T$ **do**             // backpropagation
         visits to $(\mathbf{x}_{t''}, t'')$ += 1
         reward from $(\mathbf{x}_{t''}, t'')$ += $r(\hat{\mathbf{x}}_0, \mathbf{c})$
8:          **end for**
         $\mathbf{x}_{t-1} \leftarrow \arg\max_{c_j(\mathbf{x}_t)} \frac{\text{reward from } (c_j(\mathbf{x}_t), t-1)}{\text{\# visits to } (c_j(\mathbf{x}_t), t-1)}$
9:      **end for**
10: **end for**
11: **return** $\mathbf{x}_0$

---

---

**Algorithm 5** Rejection Sampling

---

**Require:** Discretization timesteps $T$, context (e.g. class label) $\mathbf{c}$, learned denoising network $D_\theta$, number of samples $N$, sampling step function $f$, reward function $r$, initial noise sample $\mathbf{x}_T$
1: **for** $n = 1$ to $N$ **do**
2:      **for** $t = T$ to $1$ **do**
         Sample noise candidate $\mathbf{z}_t^{(n)} \sim \mathcal{N}(\mathbf{0}, \sigma_t^2 \mathbf{I})$
3:          $\mathbf{x}_{t-1}^{(n)} = f(\mathbf{x}_t, t, \mathbf{c}, \mathbf{z}_t^{(n)})$
4:      **end for**
5: **end for**
6: **return** $\mathbf{x}_0^{(\arg\max_{n\in\{1,\ldots,N\}} r(\mathbf{x}_0^{(n)}, \mathbf{c}))}$

---

---
**Algorithm 6** Beam Search
---
**Require:** Discretization timesteps $T$, context (e.g. class label) $\mathbf{c}$, learned denoising network $D_\theta$, branching factor $N$, beam width $B$, sampling step function $f$, reward function $r$, initial noise samples $\{\mathbf{x}_T^{(j)}\}_{j=1}^B$
1: **for** $t = T$ to 1 **do**
    Sample $N$ noise candidates $\{\mathbf{z}_t^{(i)}\}_{i=1}^N \sim \mathcal{N}(\mathbf{0}, \sigma_t^2 \mathbf{I})$
    $\mathcal{C} = \emptyset$
2:     **for** $j = 1$ to $B$ **do**
3:         **for** $i = 1$ to $N$ **do**
           Add $f\left(\mathbf{x}_t^{(j)}, t, \mathbf{c}, \mathbf{z}_t^{(i)}\right)$ to $\mathcal{C}$
4:         **end for**
5:     **end for**
    Let $\mathbf{x}_{t-1}^{(1\cdots B)}$ be the $B$ latents in $\mathcal{C}$ whose one-step denoised predictions $\hat{\mathbf{x}}_0^{(t-1)}$ (computed via Tweedie's formula [Efron, 2011]) have the highest scores under $r$
6: **end for**
7: **return** $\mathbf{x}_{\arg\max_{j \in \{1,\ldots,B\}} r\left(\mathbf{x}_0^{(j)}, \mathbf{c}\right)}$
---

---
**Algorithm 7** Zero-Order Search
---
**Require:** Discretization timesteps $T$, context (e.g. class label) $\mathbf{c}$, learned denoising network $D_\theta$, max. deviation scaling factor $\lambda$, number of pivot updates $K$, number of samples $N$, sampling step function $f$, reward function $r$, initial noise sample $\mathbf{x}_T$
1: **Sample** $\mathbf{p} \sim \mathcal{N}(\mathbf{0}, \sigma_t^2 \mathbf{I})$ // pivot
2: **for** $t = T$ to 1 **do**
    **Sample** $\mathbf{p} \sim \mathcal{N}(\mathbf{0}, \sigma_t^2 \mathbf{I})$
3:     **for** $k = 1$ to $K$ **do**
        Sample $N$ noise candidates $\{\mathbf{z}_t^{(i)}\}_{i=1}^N \sim \mathcal{N}(\mathbf{0}, \sigma_t^2 \mathbf{I})$, for each $i$ let $\mathbf{z}_t^{(i)} = \lambda\sqrt{2CHW}\mathbf{z}_t^{(i)} + \mathbf{p}$
        Set $\mathbf{p} = \mathbf{z}_t^{(i)}$ for $i$ s.t. one-step $\hat{\mathbf{x}}_0$ prediction using $\mathbf{z}_t^{(i)}$ attains highest score under $r$
4:     **end for**
    Set $\mathbf{x}_{t-1} = f(\mathbf{x}_t, t, \mathbf{c}, \mathbf{p})$
5: **end for**
6: **return** $\mathbf{x}_0$
---

# D $\epsilon$-greedy Regret Analysis

To strengthen our claims on $\epsilon$-greedy's superiority, we derive its regret bound, showing that—even with only black-box access to a Lipschitz surrogate—simple regret decays sublinearly in the total number of global samples.

**Theorem 1 (Regret of $\epsilon$-greedy search)** *Let the noise at diffusion step $t$ be $\mathbf{z}_t \sim \mathcal{N}(\mathbf{0}, \sigma_t^2 \mathbf{I}_d)$. Fix a confidence level $\eta$ and truncate noise-space to the high-probability ball $\Gamma_t = \left\{ \mathbf{z} \in \mathbb{R}^d : |\mathbf{z}| \leq \sigma_t\sqrt{d\ln(1/\eta)} \right\}$. Assume the surrogate reward $\tilde{R}_t : \Gamma_t \to [0, 1]$ is L-Lipschitz and satisfies*

$$\left| \tilde{R}_t - r(\mathbf{x}_0(\mathbf{z}), \mathbf{c}) \right| \leq \delta \qquad \forall \, \mathbf{z} \in \Gamma_t.$$

*Run Algorithm 1 ($\epsilon$-greedy local search) for $K$ iterations with $N$ candidate points per iteration and global-exploration probability $\epsilon$, and let the final pivot be $\mathbf{p}_t$. Then the expected simple regret at step $t$ obeys*

$$\mathbb{E}\left[\max_{\mathbf{z} \in \Gamma_t} \widetilde{R}_t(\mathbf{z}) - \widetilde{R}_t(\mathbf{p}_t)\right] \leq \delta + C_d\big(\epsilon N K\big)^{-1/d},$$

*where the constant $C_d > 0$ depends only on the dimension $d$. Summing over $T$ diffusion steps yields*

$$R_{tot} = \sum_{t=1}^T \mathbb{E}\left[\max_{\mathbf{z} \in \Gamma_t} \widetilde{R}_t(\mathbf{z}) - \widetilde{R}_t(\mathbf{p}_t)\right] \leq T\left[\delta + C_d(\epsilon N K)^{-1/d}\right].$$

Table 3: **NFE formulas by sampling method using $T$ timesteps.** Note that NFEs denote "number of function evaluations" (i.e., number of calls to $D_\theta$) for sampling a single image.

| Method | NFE formula |
|---|---|
| Naive Sampling | $T$ |
| Best of $N$ Sampling | $NT$ |
| $(N, B)$-Beam Search | $(N + B)T$ |
| $(N, S)$-MCTS | $(N + S)T^2$ |
| $(N, K)$-Zero-Order Search | $NKT$ |
| $(N, K)$-$\epsilon$-greedy | $NKT$ |

**Proof** *By Gaussian concentration, $\mathbf{z}_t$ lies in $\Gamma_t$ with probability at least $1 - \eta$, so we may restrict attention to $\Gamma_t$ at negligible cost. Each global-exploration draw is uniform on $\Gamma_t$, and in total the algorithm collects $M = \epsilon NK$ such draws. It is a well-known result that the best of $M$ uniform samples over a $d$-dimensional, Lipschitz-reward domain achieves expected simple regret on the order of $M^{-1/d}$. Since the $\epsilon$-greedy pivot never decreases in surrogate reward, its final surrogate value is at least that of the best global sample, inheriting the same $\Theta(M^{-1/d})$ rate. Finally, replacing the surrogate by the true reward adds at most $\delta$, yielding the per-step bound; summing over $T$ steps gives the stated total regret.* ∎

Note the following key points:

- **Polynomial decay in global samples.** With $M = \epsilon NK$ global draws per step, regret scales as $O(M^{-1/d})$, so increasing the total sample budget drives simple regret to zero at the classic random-search rate.

- **Bias floor $\delta$.** In practice we can take $\delta$ to be quite low as it is simply the approximation error induced by application of Tweedie's formula, $|r(\mathbf{x}_0, \mathbf{c}) - r(\hat{\mathbf{x}}_0^{(t)}, \mathbf{c})|$.

# E  Visualizing $\epsilon$-greedy Noise Search

We display a animation visualizing the $\epsilon$-greedy search process at this link. For each timestep $t$, over the $K = 20$ local search iterations, we plot the 2D t-SNE [van der Maaten and Hinton, 2008] projection of the pivot and track it over time. The animation displays visually the emergent properties of $\epsilon$-greedy we describe in subsubsection 5.1.3 of the main paper: at early and late denoising steps, the search is mostly random global exploration; but at intermediate steps, after a couple initial random draws, all subsequent iterations are local hill-climbing.

# F  Additional Experiments

## F.1  Scaling up dataset size

Below, we report EDM results using the ImageNet reward on 300 images, a $10\times$ increase in dataset size. This confirms the superiority of our $\epsilon$-greedy search.

Table 4: **EDM results by sampling method using classifier reward.** We use the same algorithm hyperparameters as in Table 1 of the main paper.

| Reward | Naive | Best-of-4 Sampling | Beam Search | MCTS | Zero-Order Search | $\epsilon$-greedy |
|---|---|---|---|---|---|---|
| Classifier | 0.3738 | 0.6413 | 0.6928 | 0.7311 | 0.6818 | **0.9790** |

## F.2 Increasing task complexity

We provide results evaluating methods on a far harder set of prompts containing compositional information Black et al. [2024], specifically those of the form "a [SUBJECT] [PHRASE]," e.g. "a lion washing the dishes." Table 5 displays these results; $\epsilon$-greedy is again superior across the board.

Table 5: **SD results by sampling method, across reward functions, on compositional prompts.** We use the same hyperparameters as in Table 2 of the main paper.

| Method | Brightness | Compressibility | CLIP |
|---|---|---|---|
| Naive Sampling | 0.4579 | 0.6866 | 0.2578 |
| Best-of-4 Sampling | 0.5257 | 0.7639 | 0.2618 |
| Beam Search | 0.5179 | 0.7690 | 0.2646 |
| MCTS | 0.5921 | 0.7717 | 0.2954 |
| Zero-Order Search | 0.4573 | 0.7128 | 0.2802 |
| $\epsilon$-greedy | **0.6312** | **0.7869** | **0.3053** |

## F.3 Increasing reward complexity

To test our method on harder rewards capturing more fine-grained details of images, we provide results evaluating each method using the counting reward from [Kim et al., 2025b], normalized to $[0, 1]$. Each prompt is a comma-separated list of 1–6 clauses, each of the form "[NUMBER] [OBJECT]," e.g. "five horses, three cars, one train, five airplanes." The counting reward function leverages open-source object detection models to compute the MSE between the actual and ground truth number of objects in the generated image, averaged across all objects described in the prompt. Even with this hard, non-differentiable counting reward function, we again see that $\epsilon$-greedy is superior across the board.

Table 6: **SD results by sampling method, using counting reward.**

| Reward | Naive | Best-of-4 | Beam Search | MCTS | Zero-Order Search | $\epsilon$-greedy |
|---|---|---|---|---|---|---|
| Counting Reward | 0.3076 | 0.3156 | 0.4318 | 0.4377 | 0.3846 | **0.5384** |

## F.4 Varying model family, reward function

To support our claim of $\epsilon$-greedy's superiority, we display results using a larger text-to-image model (SDXL [Podell et al., 2023]) and the text-to-image human preference reward model ImageReward [Xu et al., 2023a]. We use the same prompts and algorithm hyperparameters as in Table 2 of the main paper, and normalize the ImageReward score to $[0, 1]$. Results are shown in Table 7.

$\epsilon$-greedy remains the highest-performing sampling method in this experimental setting as well, highlighting its superiority across reward functions and diffusion model families/sizes.

Table 7: **SDXL results by sampling method, using ImageReward [Xu et al., 2023a] as the reward function.**

| Reward | Naive | MCTS | Zero-Order Search | $\epsilon$-greedy |
|---|---|---|---|---|
| Image-Reward | 0.4668 | 0.5152 | 0.5036 | **0.5830** |

## F.5 Tweedie estimates vs. learned reward models, few-step models

A possible concern with our proposed approach is that predicting $\mathbf{x}_0$ solely using Tweedie's formula with a base model remains too crude, potentially yielding relatively low benefits, especially during the initial sampling stages. Alternatives include using a learned soft value function on latents or a few-step model distilled from the corresponding base model, which might lead to better results.

We test this empirically, showing that using a Tweedie estimate is extremely competitive with, and often *actually outperforms* both these alternatives, across model architectures, sampling methods, and reward functions.

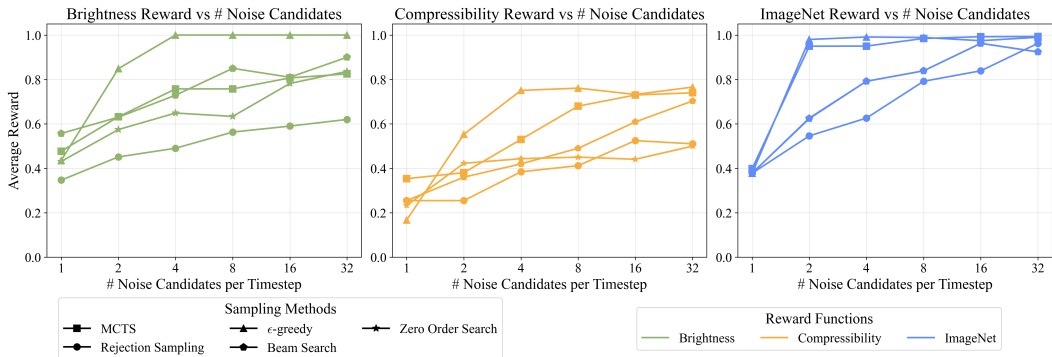

Figure 3: **Performance scaling with number of noise candidates.** Performance of sampling methods as the number of noise candidates $N$ per timestep increases, measured on three reward functions (brightness, compressibility, ImageNet). $\epsilon$-greedy achieves the best scaling law, attaining the highest rewards at the majority of $N$ values; however, it experiences non-monotonic gains due to its greedy local selection, whereas MCTS—by approximating exhaustive search—shows steady, monotonic improvement.

First, in the class-conditional image generation setting, we compare running the ImageNet classifier on the Tweedie estimate $\hat{x}_0^{(t)}$, versus running a classifier on the noisy latent $x_t$ itself at each timestep $t$. For this "noisy classifier" we adopt the ImageNet-1k $64 \times 64$ classifier from Dhariwal and Nichol [2021], trained to classify latents across timesteps and noise scales. As with the regular classifier reward, the reward here is the noisy classifier's probability assigned to the ground-truth class. Table 8 displays these results.

Table 8: **EDM results by sampling method using either a classifier reward on the Tweedie estimate, or a learned classifier on the noisy latents.**

| Reward | Beam Search | Zero-Order Search | $\epsilon$-greedy |
|---|---|---|---|
| ImageNet classifier on Tweedie-estimate | 0.9666 | **0.6261** | 0.9885 |
| Learned ImageNet classifier on latents | **0.9792** | 0.6099 | **0.9998** |

Second, in the harder text-to-image generation setting, we compare computing the CLIP reward on the Tweedie estimate, with instead computing the CLIP reward on the clean sample generated using a one-step distilled model from the current latent and noise candidate, at each timestep $t$. As in the main paper, we use Stable Diffusion v1.5, and for the one-step model use InstaFlow-0.9B [Liu et al., 2024] which was distilled from SD1.5. Results are shown in Table 9.

Table 9: **SD results by sampling method using CLIP reward either on the Tweedie estimate or the predicted clean sample using a one-step distilled model.**

| Reward | Beam Search | Zero-Order Search | $\epsilon$-greedy |
|---|---|---|---|
| Tweedie-estimate | **0.2682** | 0.2689 | **0.2973** |
| One-step model distilled from SD1.5 | 0.2671 | **0.2825** | 0.2820 |

## G  Hyperparameter Settings & Scaling

### G.1  Scaling with Increasing $N, K$

It is important to recognize that $\epsilon$**-greedy attains the best "scaling law" among different search methods empirically**, although with drawbacks compared to MCTS. Figure 3 displays the performance by sampling method as the number of noise candidates $N$ per timestep is scaled up, with all other hyperparameters fixed. Although $\epsilon$-greedy consistently achieves the best performance for sufficiently high $N$ as expected, it does face the limitation of not improving monotonically when scaling up $N$. This is because $\epsilon$-greedy (per its name) greedily selects the best noise based on $\hat{\mathbf{x}}_0^{(t)}$ at

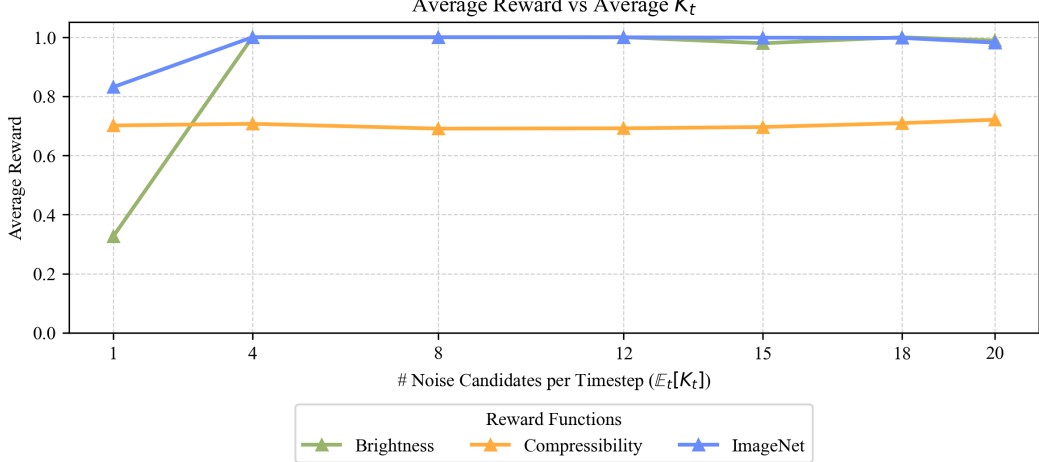

Figure 4: **EDM results by sampling method, varying $\mathbb{E}_t[K_t]$.** We use the same class labels and (other than $K$) algorithm hyperparameters as in Table 1 of the main paper.

timestep $t$; hence, as we increase the number of noise candidates per timestep, it may greedily select a better noise at the current step which ends up being a worse one globally (i.e., a locally worse noise would have led to a better overall noise trajectory). MCTS is the only method that should (and indeed does) see monotonic improvements as it approximates exhaustive search over the $N^T$ possible noise trajectories.

For this same reason, as shown in Figure 5 we see a similar lack of monotonicity in $\epsilon$-greedy and zero-order search's scaling as a function of the number of search iterations $K$ per timestep; though again $\epsilon$-greedy is empirically extremely performant, attaining higher rewards than vanilla zero-order search across $K \in [1, 64]$ (leftmost figure).

## G.2   Varying $K$ across timesteps

Recall that $\epsilon$-greedy requires $NKT$ NFEs for $K$ local search iterations. Note, however, that this assumes $K$ is fixed across timesteps. Letting $K_t$ be the number of local search iterations at timestep $t$, we can significantly bring down $\mathbb{E}_t[K_t]$ by noting that $K_t$ only need be high at intermediate denoising steps where de-mixing occurs, and can be low at extreme (beginning and end) timesteps. We report $\epsilon$-greedy performance as a function of $\mathbb{E}_t[K_t]$ below in Figure 4.

Indeed, $\epsilon$-greedy attains strong performance across all reward functions, even with extremely low average $K_t$ (can retain performance within 0.002 of the highest value with $\mathbb{E}_t[K_t]$ as low as 4), *highlighting our method's computational efficiency compared to MCTS.*

## G.3   Sweeps over $\lambda, \epsilon$

We additionally conduct hyperparameter sweeps for $\lambda \in [0, 1]$ for zero-order and $\epsilon$-greedy) search, and the $\epsilon$ parameter in $\epsilon$-greedy (bottom right 2 plots of Figure 5). We find optimal $\epsilon = 0.4$ and $\lambda \in [0.1, 0.2]$ across sampling methods and reward functions. Although the drop-offs in performance with higher $\epsilon$ and $\lambda$ (especially step for $\lambda$) seem surprising, we purport it is actually expected behavior again due to the greediness of zero-order and $\epsilon$-greedy search; the larger the $\epsilon$ and/or $\lambda$, the more we can move in "noise space" at each timestep, meaning the more likely it is for us to find a locally better but globally worse noise candidate. For large $\lambda$ especially, we may observe noise candidates far outside $\mathcal{N}(\mathbf{0}, \sigma_t^2 \mathbf{I})$ for which $\hat{\mathbf{x}}_0^{(t)}$ is no longer a meaningful sample under the reward function, which explains the very low scores in the lower right plot. (This also explains why $\epsilon$-greedy suffers less from this problem than vanilla zero-order search; it always has nonzero probability of selecting a random Normal sample and escaping these degenerate noise regions.)

Overall, the optima we find seem to be a "sweet spot" in this regime—these values are large enough to permit exploration at each timestep, but not too large as to succumb to the aforementioned behavior.

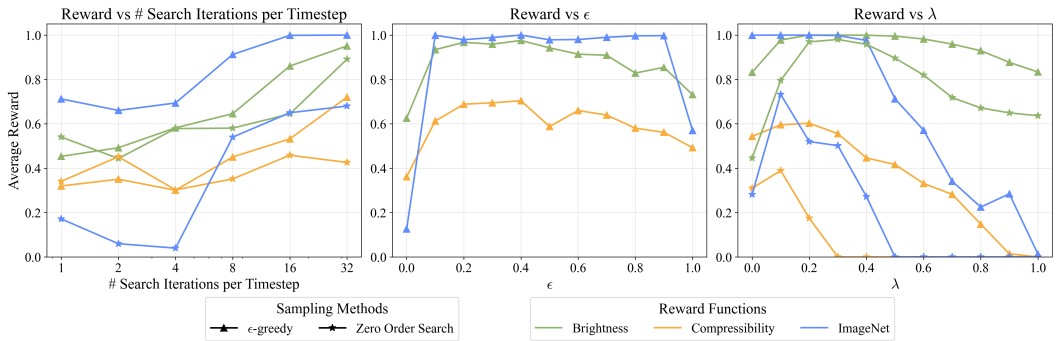

Figure 5: *(Left)* **Average reward vs. number of search iterations per timestep.** *(Center)* **Sweep over** $\epsilon$ **for** $\epsilon$**-greedy**. Highlights an optimal $\epsilon \approx 0.4$ that balances exploration and exploitation. *(Right)* **Reward vs. maximum step size scaling factor** $\lambda$ **for zero-order and** $\epsilon$**-greedy search.** Demonstrates optimal $\lambda \in [0.1, 0.2]$ and drastic performance drops at large $\lambda$, due to candidate set being overwhelmed by degenerate noise samples far outside the standard Normal distribution.

Table 10: **SD results by sampling method, varying the classifier-free guidance scale.** We evaluate only CLIP reward as the brightness and compressibility rewards are generally insensitive to classifier-free guidance.

| Method | cfg $= 1.5$ | cfg $= 3.0$ | cfg $= 7.5$ | cfg $= 9.0$ | cfg $= 12.0$ |
|---|---|---|---|---|---|
| Naive Sampling | 0.2446 | 0.2635 | 0.2457 | 0.2650 | 0.2671 |
| Best-of-4 Sampling | 0.2612 | 0.2743 | 0.2776 | 0.2783 | 0.2759 |
| Beam Search | 0.2673 | 0.2314 | 0.2682 | 0.2801 | 0.2780 |
| MCTS | 0.2828 | 0.2712 | 0.2885 | 0.2971 | 0.2827 |
| Zero-Order Search | 0.2539 | 0.2612 | 0.2689 | 0.2443 | 0.2801 |
| $\epsilon$-greedy | **0.2958** | **0.3092** | **0.2973** | **0.3055** | **0.2994** |

### G.4 Varying classifier-free guidance scale, number of timesteps $T$

See Table 10 and Table 11 above. These results confirm the *superiority of $\epsilon$-greedy irrespective of guidance scale and $T$*.

## H Qualitative Results

To assist the reader in understanding the optimization process, we offer visualizations of the generated $512 \times 512$ resolution images from Stable Diffusion using our different sampling methods. The aim is to provide a qualitative evaluation of the proposed $\epsilon$-greedy method.

As can be seen in the images below, our inference-time algorithms and most noticeably $\epsilon$-greedy indeed yield samples with significantly higher rewards, while avoiding mode collapse and ensuring distributional guarantees by virtue of being solely inference-time noise search methods.

Not only do we *not* cherry-pick examples, we in fact display results using prompts that are "hard/adversarial" with respect to the reward, i.e. "a *black* lizard" for brightness or "a potted fern *with*

Table 11: **EDM results by sampling method, varying the value of $T$.** Each cell contains two values, corresponding to scores under the Brightness / Compressibility reward functions.

| Method | $T = 18$ | $T = 32$ | $T = 64$ | $T = 128$ | $T = 256$ |
|---|---|---|---|---|---|
| Naive Sampling | 0.4965/0.3563 | 0.3912/0.3921 | 0.4621/0.4549 | 0.4849/0.4052 | 0.4541/0.4250 |
| Best-of-4 Sampling | 0.5767/0.4220 | 0.5666/0.4992 | 0.6475/0.5084 | 0.5801/0.5101 | 0.5923/0.4917 |
| Beam Search | 0.6334/0.4679 | 0.6013/0.4321 | 0.5742/0.5023 | 0.6162/0.4517 | 0.6618/0.4879 |
| MCTS | 0.7575/0.5395 | 0.6910/0.5153 | 0.6831/0.5624 | 0.6957/0.5018 | 0.7494/0.5789 |
| Zero-Order Search | 0.6083/0.3751 | 0.9999/0.7417 | 0.9999/**0.7607** | 0.9999/0.7671 | 0.9999/**0.7677** |
| $\epsilon$-greedy | **0.9813**/**0.7208** | **0.9999**/**0.7424** | **0.9999**/0.7588 | **0.9999**/**0.7680** | **0.9999**/0.7672 |

*hundreds of leaves*" for compressibility. The success of our method even on these samples highlights two desiderata our algorithms satisfy: **faithfulness to prompt** and **lack of mode collapse**.

Quite remarkably, we see that when using our sampling algorithms, the denoising process consistently uses the additional NFEs to employ one of a *limited* number of reward optimization "strategies" shared across prompts and methods. Some examples of these common strategies, prefixed by the reward functions they apply to, include:

- (compressibility) blurring content;
- (brightness, compressibility) occluding or minimizing the subject of the image;
- (compressibility) changing style from photorealistic to "animated/cartoonish";
- and (CLIP) emphasizing distinctive features of the image subject. Examples (see Figure 8) include the curved beak and wide brown region around the eyes of the kākāpō and the unique red nose of the red wolf. It's particularly remarkable that just noise optimization alone for the CLIP reward increases faithfulness to elements of the prompt like "*three* **rockhoppers in a line**;" following detailed prompts like these is a notorious and long-standing problem in diffusion [Black et al., 2024, Kim et al., 2025a], and *none of these features together were captured by the other sampling methods besides $\epsilon$-greedy*.

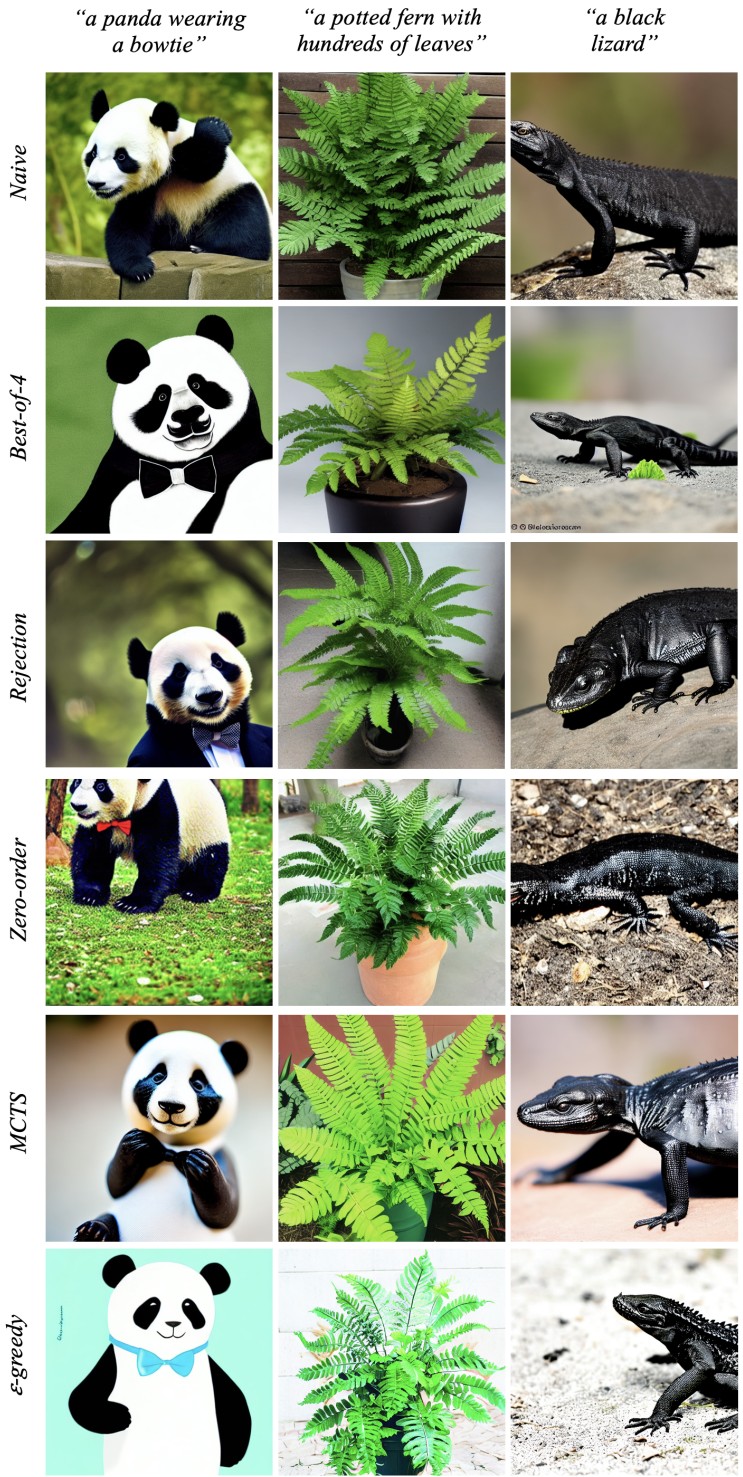

Figure 6: Visualizations of images generated with Stable Diffusion v1.5, with varying sampling strategies, using the brightness reward function.

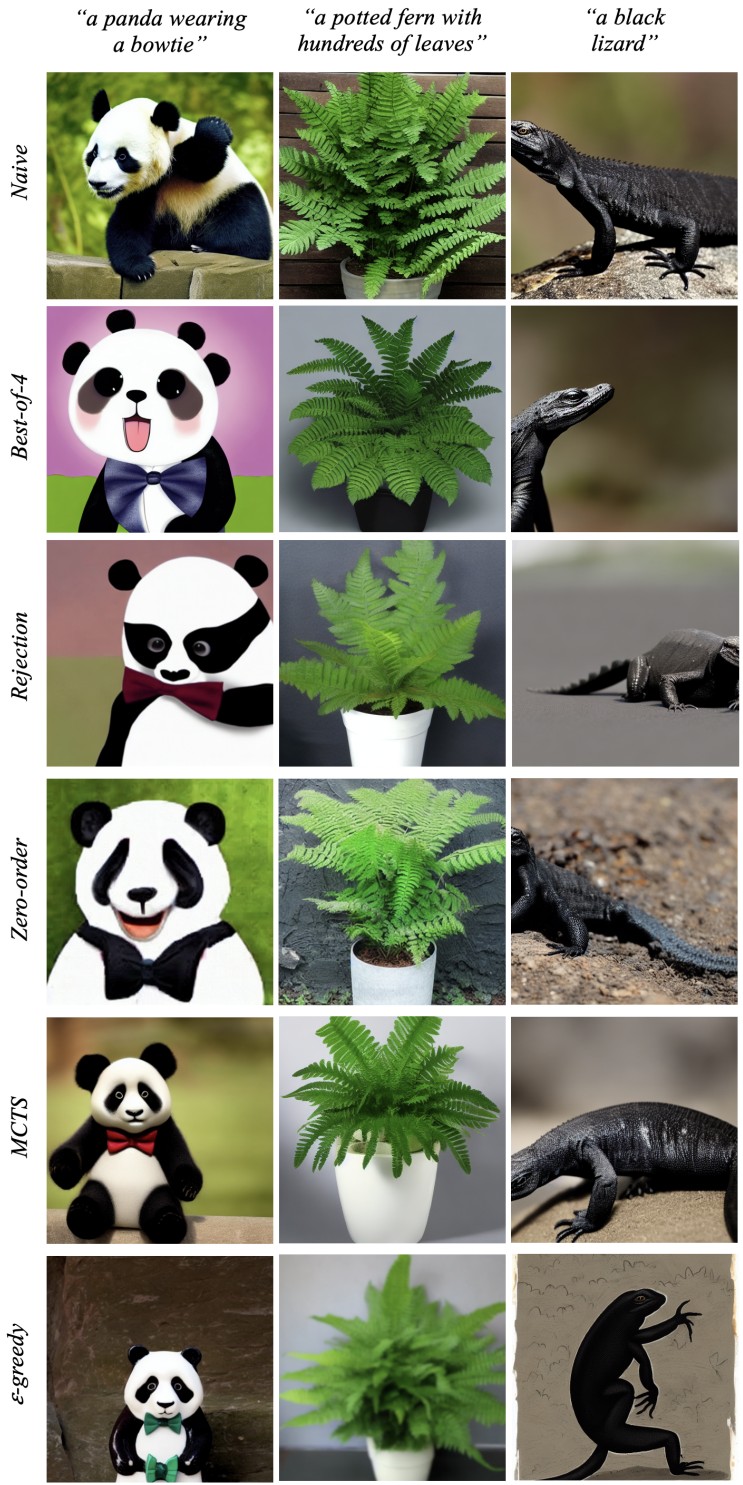

Figure 7: Visualizations of images generated with Stable Diffusion v1.5, with varying sampling strategies, using the compressibility reward function.

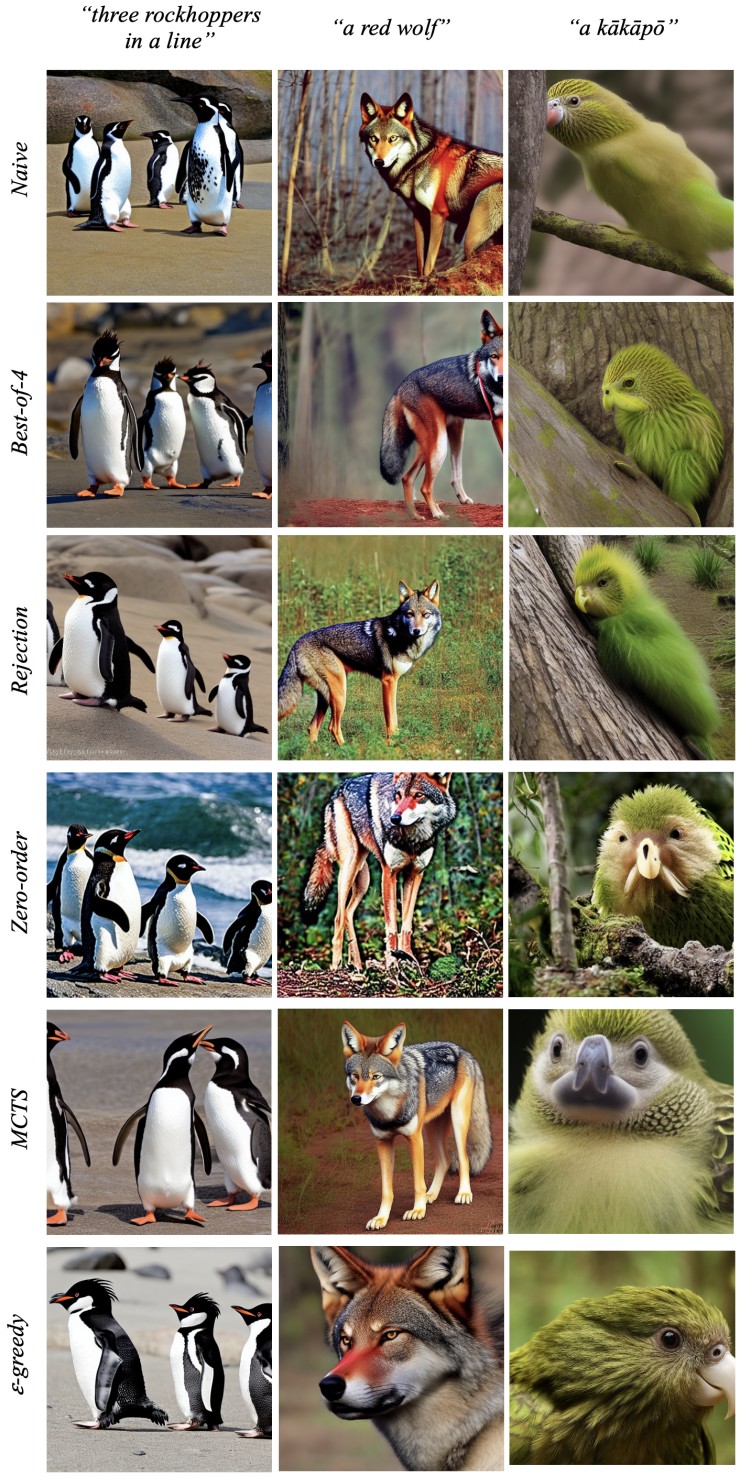

Figure 8: Visualizations of images generated with Stable Diffusion v1.5, with varying sampling strategies, using the CLIP reward function.

# I  Broader Impacts

Our inference-time optimization framework for diffusion models empowers users to steer generative outputs toward arbitrary, user-defined rewards. In positive scenarios, artists and designers can interactively refine visual concepts in real time, accelerating creative workflows; educators and accessibility advocates can tune outputs for enhanced contrast or simplified imagery to aid visually impaired learners; and scientists can optimize for perceptual clarity or domain-specific criteria (e.g. highlighting anatomical structures in medical images), improving interpretability of complex data visualizations.

However, the same mechanism that makes our method so flexible also amplifies risks. By optimizing for malicious objectives—such as photorealistic deepfakes tailored to evade detection or synthetic imagery crafted to reinforce biased or harmful stereotypes—adversaries could undermine trust in visual media, facilitate disinformation campaigns, or violate individual privacy. Moreover, if reward functions inadvertently encode cultural or demographic biases, optimized outputs may perpetuate unfair or exclusionary representations.

Notably, however, the proposed methods are simply algorithms with which to sample from existing, already released diffusion models; hence safeguards paired with those models remain in place. For instance, models like those in the Stable Diffusion family are paired with NSFW content checkers that will filter out images with harmful content that might be generated with our algorithms, helping mitigate harm from cases where users try to optimize objectives that reward for such content.

# J  Discussion and Next Steps

While our method yields state-of-the-art class-conditional and text-to-image generation results, it is not without its limitations:

## J.1  Compute Efficiency

For $N$ noise candidates and $K$ local search iterations per timestep, the $\epsilon$-greedy approach requires $NK$ times the NFEs compared to vanilla sampling for generating a single image. While this approach remains computationally linear with respect to the number of timesteps $T$, thus making it significantly less costly than more computationally intensive alternatives like MCTS, it nonetheless poses a computational burden. This additional computational cost could limit practical deployment scenarios, particularly where resources or latency constraints are critical factors.

To mitigate this, recalling Figure 2, as a proof-of-concept, we run zero-order search and $\epsilon$-greedy with $K = 20$ for only $\{t : 0.01 \le \sigma_t \le 1\}$, $K = 1$ otherwise. This yields rewards within $\pm 0.04$ of the original results (where $K = 20$ is used at all denoising steps), but cuts the NFEs by more than half. We recognize exploration of similar approaches to increase computational efficiency as an important direction for future research.

## J.2  Population-Based Metrics

Unlike traditional population-based metrics such as FID or Inception Score [Salimans et al., 2016], our evaluation protocol focused on single-image rewards (e.g., classifier confidence, brightness, compressibility). Evaluating on population-based metrics is part of our future work. However, it is worth noting that the type of single-objective optimization we employ has become somewhat of a standard practice in recent works regarding reward optimization of diffusion models; e.g. Yeh et al. [2025], Black et al. [2024] use the exact same experimental setup as we do. In addition, our method poses no threat to the population-based guarantees that measures like FID try to make, since all our methods—by virtue of being inference-time search algorithms that are completely gradient-free—avoid mode collapse and ensure distributional guarantees [Yeh et al., 2025].

## J.3  Reward Overoptimization

For high enough $\lambda$, the zero-order and $\epsilon$-greedy searches occasionally over-optimize the brightness and compressibility rewards, generating full-white and single-color images, respectively. This is due to the fact that, for large enough $\lambda$, the local-search iterations enable finding noises potentially far

outside $\mathcal{N}(\mathbf{0}, \sigma_t^2 \mathbf{I})$ that no longer remain faithful to the context $\mathbf{c}$. While (1) this only occurs with rewards $r$ that grade samples $\mathbf{x}_0$ independent of their corresponding contexts $\mathbf{c}$, indicating that such over-optimization could be construed as extreme efficacy of our method; and (2) this behavior can be easily mitigated by reducing/tuning $\lambda$; alterations to this method to prevent over-optimization irrespective of hyperparameter settings remains an important direction of future work. Examples include the regularization mechanisms proposed by Zhang et al. [2024], Tang et al. [2024].

### J.4 Future Work

Directions for future work include:

**Reducing computational overhead.** Future work should aim to lower the cost of per-step search, for instance through adaptive scheduling of NFEs or by learning proposal distributions that make candidate selection more efficient.

**Exploring alternative bandit solvers.** Hybrid methods, such as combining tree-based search with bandit algorithms, may improve sample efficiency and accelerate convergence.

**Mitigating reward overoptimization.** Regularization strategies could help reduce overfitting to specific reward signals.

**Modeling diffusion-step interactions.** More sophisticated search approaches should account for correlations between noise candidates across timesteps, rather than treating each step independently.

**Learning compute-optimal regions.** Identifying points during denoising where search is most valuable could allow resources to be allocated selectively, improving efficiency without sacrificing quality.

**Using reward ensembles.** Averaging across multiple reward metrics, such as different VLM-based signals, can provide more stable guidance than relying on a single objective.

**Human-in-the-loop extensions.** Future systems could be adapted to operate in preference-based settings where only human ratings or feedback are available instead of closed-form reward functions.

