# OpenReview forum: "Test-Time Scaling of Diffusion Models via Noise Trajectory Search"
_NeurIPS.cc/2025/Conference — NeurIPS 2025 poster_

### Official Review · Reviewer_D5sF · 2025-06-20

**Clarity:** 3
**Significance:** 3
**Originality:** 4
**Rating:** 6
**Confidence:** 4

**Summary:**

This is a quite interesting and meaningful work, presenting a novel test-time scaling algorithm for diffusion models. The core idea is to treat each sampling step as an independent contextual bandit problem and then use an ϵ-greedy search algorithm to achieve a more comprehensive search. Compared to some previous test-time scaling algorithms, it is remarkably efficient and performs exceptionally well.

**Questions:**

No.

**Ethical Concerns:**

["NO or VERY MINOR ethics concerns only"]

**Final Justification:**

Since the author added new experiments that resolved my concerns, I have raised my score to "strong accept."

**Limitations:**

Yes.

**Paper Formatting Concerns:**

No.

**Quality:**

3

**Strengths And Weaknesses:**

**Strength**:

1. The problem of test-time scaling in diffusion models is notably challenging. The method proposed by the authors is highly novel, and I personally believe its exceptional performance stems from the $\epsilon$-greedy search algorithm, which ensures that each sampling action is based on the previous iteration, thereby achieving a "cumulative gain" effect.

2. The authors' proposed method is simple and efficient, which does not detract from its novelty or its beneficial impact on other works. For instance, this test-time scaling algorithm might enable step distillation algorithms for diffusion models to achieve superior results.

3. The writing and logic of the article is clear and very engaging for the reader.

**Weakness**:

I believe this paper has two main shortcomings.

1. The experiments in the paper are relatively limited. The authors might consider conducting further experiments on SDXL, SD3.5, and FLUX, evaluating them on benchmarks such as HPS v2, Image-Reward, GenEval, and T2I-CompBench.

2. The approach of predicting $x_0$ solely using Tweedie's formula with a base model remains too crude, potentially yielding relatively low benefits, especially during the initial sampling stages. The authors could consider using a few-step model distilled from the corresponding base model to execute Tweedie's formula, which might lead to better results.

---

> ### Author Rebuttal · Authors · 2025-07-31
>
> We thank the reviewer for their feedback, and address their comments below:
>
> 1. > The experiments in the paper are relatively limited. The authors might consider conducting further experiments on SDXL, SD3.5, and FLUX, evaluating them on benchmarks such as HPS v2, Image-Reward, GenEval, and T2I-CompBench.
>
> **We have included a suite of additional experiments that more comprehensively test our proposed method**:
> - Evaluate SD across classifier-free guidance scale values **(Response to Reviewer ACK2, Table R2)**
> - Evaluate EDM across # of denoising steps **(Response to Reviewer ACK2, Table R3)**
> - Evaluate EDM on 300 images, a 10x increase in dataset size **(Response to Reviewer Sj6K, Table R4)**
> - Evaluate SD on harder prompts containing compositional information **(Response to Reviewer Sj6K, Table R5)**
> - Evaluate SDXL [1] on Image-Reward [2] **(Table R6 below)**
>
> | **Reward**           | **Naive** |  **MCTS** | **Zero-Order Search** | **$\epsilon$-greedy** |
> | -------------------- | --------- |  -------- | --------------------- | --------------------- |
> | **Image-Reward** [2] | 0.4668      | 0.5152     | 0.5036                  | **0.5830**                  |
>
> ***Table R6: SDXL results by sampling method, using Image-Reward [2] as the reward function**. We use the same prompts and algorithm hyperparameters as in Table 2 of the main paper. We normalize Image-Reward score to $[0,1]$.*
>
> **$\epsilon$-greedy remains the highest-performing sampling method across all these experimental settings, highlighting its superiority across reward functions and diffusion model families/sizes.**
>
>
> 2. > The approach of predicting x_0 solely using Tweedie's formula with a base model remains too crude, potentially yielding relatively low benefits, especially during the initial sampling stages. The authors could consider using a few-step model distilled from the corresponding base model to execute Tweedie's formula, which might lead to better results.
>
> Past work has shown that computing reward on the Tweedie-estimate $\mathbb{E}[\mathbf{x}_0 \mid \mathbf{x}_t, \mathbf{z}_t, \mathbf{c}]$ **actually outperforms** using learned soft value functions or few-step models. [3] compares the performance of SVDD-MC (beam search using a learned value function) and SVDD-PM (beam search using reward computed on $\hat{x}_0$ estimated via Tweedie's formula). They find that SVDD-PM is generally more robust since it does not require additional learning (i.e., it directly utilizes the ground truth reward feedback); in fact, SVDD-PM outperforms SVDD-MC across the majority of domains and experimental settings.
>
> That being said, while learning value functions (e.g. with distillation or even Q-learning algorithms such as DDPG) is orthogonal to our current work (which focuses on choice of *search algorithm* rather than per-step reward function), we are excited by this direction and consider it important future work.
>
>
> **We hope that with the provided responses & additional strong results, you consider raising your score to support strong acceptance. We are happy to address any additional questions.**
>
> [1] Podell et al., SDXL: Improving Latent Diffusion Models for High-Resolution Image Synthesis, 2023
>
> [2] Xu et al., ImageReward: Learning and Evaluating Human Preferences for Text-to-Image Generation, 2023
>
> [3] Li et al., Derivative-Free Guidance in Continuous and Discrete Diffusion Models with Soft Value-Based Decoding, 2024

---

> > ### Comment · Reviewer_D5sF · 2025-08-04
> > **Thanks for your explanation**
> >
> > I appreciate the author's thorough response, particularly the successful resolution of Weakness 1. However, I remain unconvinced by the author's justification for employing a few-step model for the Tweedie estimate, which relies solely on past experience (Weakness 2). While the author's algorithm may offer similar acceleration benefits, I believe empirical evidence is necessary to substantiate this choice. Therefore, I maintain my original score.

---

> > > ### Author Response · Authors · 2025-08-05
> > > **Requested results**
> > >
> > > We thank the reviewer for their prompt response. To augment the justification in our previous comment, we provide the results of two experiments below which show that **computing reward on the Tweedie estimate is extremely competitive with using learned reward models on latents or few-step distilled models, with the two techniques $\leq 0.02$ reward apart across model architectures, sampling methods, and reward functions.**
> > >
> > > 1. First, in the class-conditional image generation setting, we compare running the ImageNet classifier on the Tweedie estimate $\hat{\mathbf{x}}_0^{(t)}$, with instead running a classifier on the noisy latent $\mathbf{x}_t$ itself, at each timestep $t$. For this "noisy classifier" we adopt the ImageNet-1k $64 \times 64$ classifier from [1], which was trained to classify latents across diffusion timesteps and noise scales. As with the regular classifier reward, the reward here is the noisy classifier’s probability assigned to the ground-truth class.
> > >
> > > | **Reward**                                  | **Beam Search** | **Zero-Order Search** | **$\epsilon$-greedy** |
> > > | ------------------------------------------- | --------------- | --------------------- | --------------------- |
> > > | **ImageNet classifier on Tweedie-estimate** | 0.9666          | **0.6261**               | 0.9885                |
> > > | **Learned ImageNet classifier on latents**  |  **0.9792**         |    0.6099             |    **0.9998**        |
> > >
> > > ***Table R7: EDM results by sampling method using either a classifier reward on the Tweedie estimate, or a learned classifier on the noisy latents themselves.** We use the same algorithm hyperparameters as in Table 1 of the main paper. Note that we do not report results for the naive, rejection, or MCTS sampling methods because these all compute reward on the final clean sample, hence there would be no difference between the two rewards tested in this experiment.*
> > >
> > > 2. Second, in the harder text-to-image generation setting, we compare computing the CLIP reward on the Tweedie estimate, with instead computing the CLIP reward on the clean sample generated using a one-step distilled model from the current latent and noise candidate, at each timestep $t$. As in the main paper we use Stable Diffusion v1.5, and for the one-step model use InstaFlow-0.9B [2] which was distilled from Stable Diffusion v1.5.
> > >
> > > | **Reward**                              | **Beam Search** | **Zero-Order Search** | **$\epsilon$-greedy** |
> > > | --------------------------------------- | --------------- | --------------------- | --------------------- |
> > > | **Tweedie-estimate**                    | **0.2682**         | 0.2689                | **0.2973**            |
> > > | **One-step model distilled from SD1.5** | 0.2671                | **0.2825**            | 0.2820                |
> > >
> > > ***Table R8: SD results by sampling method using CLIP reward either on the Tweedie estimate or the predicted clean sample using a one-step distilled model.** We use the same algorithm hyperparameters as in Table 2 of the main paper. We again do not report results for the naive, rejection, or MCTS sampling methods.*
> > >
> > > **We hope that with the strong results from these requested experiments, you consider raising your score to support strong acceptance!**
> > >
> > > [1] Dhariwal & Nichol, Diffusion Models Beat GANs on Image Synthesis, 2021.
> > >
> > > [2] Liu et al., InstaFlow: One Step is Enough for High-Quality Diffusion-Based Text-to-Image Generation, 2023.

---

> > > > ### Comment · Reviewer_D5sF · 2025-08-05
> > > >
> > > > No problem!

---

### Official Review · Reviewer_Sj6K · 2025-06-28

**Clarity:** 1
**Significance:** 1
**Originality:** 1
**Rating:** 4
**Confidence:** 4

**Summary:**

This paper frames test time scaling of diffusion as Markov Decision Process (MDP) with a terminal reward. To avoid excessive computation costs of tree-search methods, they view denoising as a sequence of independent contextual bandits. Then, they introduce $\epsilon$-greedy search algorithm.

**Questions:**

1. Refer to weakness above.
2. How many images are used for evaluation on Stable Diffusion?
3. I suggest to show qualitative results in the main paper.

**Ethical Concerns:**

["NO or VERY MINOR ethics concerns only"]

**Final Justification:**

The major concerns I have at the first review was
1. lack of novelty and justification.
2. experiments that are conducted on too small set.
3. experiments that are conducted on too easy task.
4. no discussion regarding the side effect.

1 is fully resolved, as the concern originates from misunderstanding.
2 is partly resolved as they have conducted experiments on the 300 images, which is statistically meaningful.
3 is resolved.
4 is partly resolved as they explain qualitative result in text and suggest solution for it.

**Limitations:**

1. Refer to weakness above.

**Paper Formatting Concerns:**

None.

**Quality:**

1

**Strengths And Weaknesses:**

Strengths
1. The paper tackles important problem, test-time scaling of diffusion models.

Weakness
1. Lack of novelty
	- The only novel component is $\epsilon$-greedy noise search.
	- SVDD [1] has already proposed method to deal with non-differentiable rewards unlike the last sentence of the abstract.
2. Lack of discussion on the proposed method.
	- The paper does not discuss why $\epsilon$-greedy method improves the sampling.
3. Unclear explanation of overall sampling process
	- In Algorithm 1, there is no explanation about how the denoising network $D_\theta$ is used. It seems the written algorithm is incomplete.
4. Limited and unconvincing experimental support
	- Insufficient scale: The valuation on ImageNet is based on only 36 images, which is too small sample size to support or claim meaningful conclusion.
	- Oversimplified tasks: The text prompts used for evaluation are very simple. To demonstrate the true value of a test-time scaling method, the experiments should involve more complex prompts which includes compositional information.
	- Quality degradation: The qualitative results (Example 2 of Figure 5, Example 3 of Figure 6) suggest, the proposed method may degrade image quality like color saturation. The authors should discuss regarding tradeoff between sample fidelity and text alignment. I suggest authors to include quantitative metrics regarding this. (measuring FID or IS along with Brightness/ Compressibility/ Classifier)
[1] Li et al., Derivative-Free Guidance in Continuous and Discrete Diffusion Models with Soft Value-Based Decoding,

---

> ### Author Rebuttal · Authors · 2025-07-31
>
> We thank the reviewer for their feedback, and address their comments below:
>
> 1. > Lack of novelty.
> The only novel component is eps-greedy noise search.
> SVDD has already proposed method to deal with non-differentiable rewards unlike the last sentence of the abstract.
>
> We would like to respectfully push back against the reviewer's statement re. lack of novelty; we outline our novel contributions below:
>
> - We are the first to investigate sophisticated step-level search algorithms (e.g. MCTS) for denoising.
> - In claiming that this is the first **practical** method for test-time noise trajectory optimization of arbitrary (non-differentiable) rewards, we mean that we are the first to **outperform MCTS** (perhaps the most generally performant tree search algorithm across domains) and **do so with compute-efficiency** ($\epsilon$-greedy requires $O(T)$, not $O(T^2)$, NFEs). $\epsilon$-greedy also achieves the **best scaling laws empirically across all sampling methods** (see Appendix D). SVDD [1] does not perform nearly as well; in fact, SVDD-PM is formally equivalent to beam search with beam width 1, and we show $\epsilon$-greedy beats a **strictly more performant** variant of SVDD-PM (namely, beam search with higher width).
> - Ours is the first work to demonstrate the importance of *adapting the search strategy – local vs. global – based on the diffusion timestep.*
> - To expand further on the comparison to SVDD [1]: it's worth distinguishing between choice of *reward* and choice of *search algorithm*. We focus our work on developing a highly-performant and compute-efficient **search algorithm** for finding the optimal noise trajectory given a reward. Hence, in our experimental setups, we **fix** a downstream reward function $r$ (one of brightness/compressibility/classifier); **fix** a per-step reward function $\tilde{R}_t = r(\hat{\mathbf{x}}_0^{(t)})$; **and then vary** the choice of search algorithm (rejection sampling, beam search, MCTS, $\epsilon$-greedy). Indeed, the SVDD search algorithm is beam search (*SVDD-PM is identical to beam search with a Tweedie estimate-based reward*), and **$\epsilon$-greedy outperforms this baseline across the board.** We treat learning a value function as in SVDD-MC to be orthogonal to choice of search algorithm; we would indeed be interested in testing $\epsilon$-greedy with soft value functions, and while this is outside our paper's scope, we consider this an important direction of future work.
>
> 2. > The paper does not discuss why eps-greedy method improves the sampling.
>
> Please see Section 5.1.3 of the paper, specifically the subsections "Leaving the denoising tree" & "Explore globally and exploit locally" - both of which provide intuitions about why $\epsilon$-greedy improves sampling. We provide a short summary of these sections below.
>
> - **Leaving the denoising tree.** Rejection sampling, beam search, and MCTS all optimize a fixed denoising tree, i.e. at each timestep $t$ we cannot explore “noise space” beyond the $N$ initially determined candidates. Zero-order search and $\epsilon$-greedy address this, with the amount of exploration in noise space dictated by the number of local search iterations $K$, the maximum step size scaling factor $\lambda$, and the global exploration factor $\epsilon$.
> - **Explore globally and exploit locally.** At each time $t$, conditioned on latent $\mathbf{x}_t$, there is a specific “region” in the noise space (call it $\mathcal{Z}$) containing noise realizations that best fit the downstream objective. In particular, $\mathcal{Z}$ is not necessarily a subset of $\mathcal{N}(0, \sigma_t^2\mathbf{I})$. If our initial pivot is far from $\mathcal{Z}$, methods such as zero-order search only allow for local exploration, i.e. we can move at most distance $K\lambda \sqrt{d/2}$ away from the pivot over $K$ local search iterations; but with $\epsilon$-greedy, we can explore the entire Normal with $\epsilon$ probability; a few such “random jumps” can move us into $\mathcal{Z}$, at which point we see large returns from hill-climbing. Furthermore, we empirically confirm that the reward landscape as a function of $\mathbf{x}_t$ is much smoother at extreme time steps rather than intermediate ones, hence there is the most performance gain from hill climbing at intermediate steps which $\epsilon$-greedy alone can exploit.
>
> Please also see **response #3 to Reviewer ACK2**, which derives the $\epsilon$-greedy regret bound and shows that—even with only black‑box access to a Lipschitz surrogate—simple regret decays *sublinearly in the total number of global samples*. Other sampling methods such as zero-order search *cannot attain this regret bound*.
>
> 3. > In Algorithm 1, there is no explanation about how the denoising network is used. It seems the written algorithm is incomplete.
>
> In the main paper, we define the function $f$ as one step of the chosen sampler (e.g. lines 2-6 of Alg. 2 for EDM, lines 2-3 of Alg. 3 for DDIM - see Appendix A in `appendix.pdf` in the supplementary material zipfile). A call to $f$ thus involves a forward pass through the denoising network; and $f$ is indeed called in line 3 of Alg. 1.
>
> We agree however that this could be explained more clearly, and will surely do so in the final paper if accepted.
>
> 4. > Insufficient scale: The valuation on ImageNet is based on only 36 images, which is too small sample size to support or claim meaningful conclusion.
>
> Below, we report EDM results using the ImageNet reward on 300 images, **a 10x increase in dataset size**. This **confirms the superiority** of our $\epsilon$-greedy search:
>
> | **Reward**           | **Naive** | **Best-of-$4$ Sampling** | **Beam Search** | **MCTS** | **Zero-Order Search** | **$\epsilon$-greedy** |
> | -------------------- | --------- | ------------------------ | --------------- | -------- | --------------------- | --------------------- |
> | **Classifier** |       0.3738                     | 0.6413           |  0.6928   |  0.7311                 |          0.6818     |**0.9790**
>
> ***Table R4: EDM results by sampling method using classifier reward**. We use the same algorithm hyperparameters as in Table 1 of the main paper.*
>
> 5. > Quality degradation: ...the proposed method may degrade image quality like color saturation. The authors should discuss regarding tradeoff between sample fidelity and text alignment. I suggest authors to include quantitative metrics regarding this. (measuring FID or IS)
>
> Unfortunately, we did not have enough compute to generate the $50K$ images recommended for accurate FID computation. However, a qualitative study revealed that for high enough $\lambda$, the zero-order and $\epsilon$-greedy searches occasionally over-optimize the brightness and compressibility rewards, producing the examples cited by the reviewer. This is due to the fact that, for large enough $\lambda$, the local-search iterations enable finding noises potentially far outside $\mathcal{N}(\mathbf{0}, \sigma_t^2\mathbf{I})$ that no longer remain faithful to the context $\mathbf{c}$. While (1) this only occurs with rewards that grade samples independent of their corresponding contexts, indicating that such over-optimization could be construed as extreme efficacy of our method; and (2) this behavior can be easily mitigated by reducing/tuning $\lambda$; alterations to this method to prevent over-optimization irrespective of hyperparameter settings remains an important direction of future work. E.g. in followup work we hope to incorporate the regularization mechanisms proposed by [2], etc.
>
> 5. > Oversimplified tasks: The text prompts used for evaluation are very simple. To demonstrate the true value of a test-time scaling method, the experiments should involve more complex prompts which includes compositional information.
>
> Below, we provide results evaluating each of our methods on **prompts with compositional information**. Following [3], each prompt is of form "a [SUBJECT] [PHRASE]," e.g. "a lion washing the dishes." **$\epsilon$-greedy is again superior across the board:**
>
> | Method                   | Brightness | Compressibility | CLIP       |
> | ------------------------ | ---------- | --------------- | ---------- |
> | **Naive Sampling**       | 0.4579     | 0.6866          | 0.2578     |
> | **Best-of-$4$ Sampling** | 0.5257     | 0.7639          | 0.2618     |
> | **Beam Search**          | 0.5179     | 0.7690          | 0.2646     |
> | **MCTS**                 | 0.5921     | 0.7717          | 0.2954     |
> | **Zero-Order Search**    | 0.4573     | 0.7128          | 0.2802     |
> | **$\epsilon$-greedy**    | **0.6312** | **0.7869**      | **0.3053** |
>
> ***Table R5: SD results by sampling method, across reward functions, on compositional prompts**. We use the same hyperparameters as in Table 2 of the main paper.*
>
> 6. > How many images are used for evaluation on Stable Diffusion?
>
> Following [4], we used natural language versions of the same 36 class labels used in EDM generation for the Stable Diffusion experiments. We plan to report results on a **10x larger** set of 300 prompts very soon within the reviewer-author discussion period.
>
> 7. > I suggest showing qualitative results in the main paper.
>
> We appreciate the suggestion, and will definitely add qualitative results to the main paper if accepted.
>
> **We hope that with the provided responses & additional strong results, you consider raising your score to support clear acceptance. We are happy to address any additional questions.**
>
> [1] Li et al., Derivative-Free Guidance in Continuous and Discrete Diffusion Models with Soft Value-Based Decoding, 2024
>
> [2] Zhang et al., Confronting reward overoptimization for diffusion models: A perspective of inductive and primacy biases, 2024
>
> [3] Black et al., Training Diffusion Models with Reinforcement Learning, 2023
>
> [4] Ma et al., Inference-Time Scaling for Diffusion Models beyond Scaling Denoising Steps, 2025

---

> ### Comment · Reviewer_Sj6K · 2025-08-03
>
> **Response 1,2,3**
>
> The comments addressed my misunderstanding and concerns.
>
> **Response 4**
>
> The expanded experiment has addressed my concerns.
>
> **Response 5**
>
> Thank authors for clarifying the failure cases and corresponding solutions.
>
> **Response 5-2**
>
> While I thank authors for the additional experiments, but it would be better if authors confirm that the method can be applied to the complicated, non-differentiable reward like counting. You may include some task in [1].
>
> [1] Kim et al., Inference-Time Scaling for Flow Models via Stochastic Generation and Rollover Budget Forcing.
>
> **Response 6, 7**
>
> Thank authors for the response.

---

> ### Author Response · Authors · 2025-08-05
> **Requested results**
>
> We thank the reviewer for their prompt response. First, we want to note that the compressibility reward already included in the main paper is indeed non-differentiable. But below, we provide results evaluating each of our methods using the exact same counting reward from [1], normalized to the range $[0,1]$. Following [1], each prompt is a comma-separated list of 1-6 clauses, each clause of form "[NUMBER] [OBJECT]" - for example, "Five horses, three cars, one train, five airplanes." **Even for the hard, non-differentiable counting reward function, we again see that $\epsilon$-greedy is superior across the board:**
>
> |                     | **Naive** | **Best-of-4** | **Beam Search** | **MCTS** | **Zero-Order Search** | **$\epsilon$-greedy** |
> | ------------------- | --------- | ------------- | --------------- | -------- | --------------------- | --------------------- |
> | **Counting Reward** | 0.3076    | 0.3156        | 0.4318          | 0.4377   |       0.3846               |       **0.5384**                 |
>
> We hope that with this additional strong result as requested, you consider raising your score to support strong acceptance!
>
> [1] Kim et al., Inference-Time Scaling for Flow Models via Stochastic Generation and Rollover Budget Forcing, 2025.

---

> > ### Comment · Reviewer_Sj6K · 2025-08-06
> >
> > Thanks for the additional experiments and this clearly addresses my concerns.

---

### Official Review · Reviewer_NHkD · 2025-06-30

**Clarity:** 3
**Significance:** 3
**Originality:** 3
**Rating:** 5
**Confidence:** 2

**Summary:**

This work propose an efficient test-time ω-greedy search algorithm, inspired by the contextual bandit problem, for optimizing noisy trajectories in diffusion models under non-differentiable reward constraints  and for efficiently optimizing noisy trajectories in diffusion models by formulating denoising as a contextual bandit problem.  It empirically outperforms Monte Carlo Tree Search  and demonstrates global adaptive exploration and local exploitation  behaviors associated with specific diffusion steps, especially being more localized in intermediate steps involving demixing. This method achieves state-of-the-art performance in image generation and is also the first practical method to optimize non-differentiable rewards during test-time sampling.

**Questions:**

Refer to weaknesses part.

**Ethical Concerns:**

["NO or VERY MINOR ethics concerns only"]

**Final Justification:**

Thanks for your reponse! I maintain my rating.

**Limitations:**

Yes

**Quality:**

3

**Strengths And Weaknesses:**

This study proposes an efficient test-time ω-greedy search algorithm inspired by contextual bandits for optimizing noisy trajectories in diffusion models under non-differentiable reward constraints. The algorithm empirically outperforms Monte Carlo Tree Search and demonstrates adaptive exploration  and local exploitation behaviors associated with specific diffusion steps, especially in the intermediate steps involving de-mixing. Through extensive comparisons with other search strategies, this study quantitatively demonstrates the critical importance of step-size adaptive search locality and shows the effectiveness of the method on EDM and stable diffusion, achieving sample quality improvement on various non-differentiable rewards such as classifier probability and VLM alignment. ​

Weaknesses: While I am not very familiar with this specific research area, the authors' presentation and experimental results seem methodologically sound and logically consistent. However, I was unable to find broader implications, limitations, and future work in the appendix. In addition, the authors only provide a small number of baselines to verify the effectiveness of the proposed testing-time strategy. I would like to know the impact of the authors' algorithm on a wider range of baselines.

---

> ### Author Rebuttal · Authors · 2025-07-31
>
> We thank the reviewer for their feedback, and address their comments below:
>
> 1. > ...I was unable to find broader implications, limitations, and future work in the appendix
>
> Please see Appendices E & F for a comprehensive discussion of broader impacts, limitations, & future work. *Note that the appendix is in `appendix.pdf` in the supplementary material zipfile.*
>
> 2. > In addition, the authors only provide a small number of baselines to verify the effectiveness of the proposed testing-time strategy. I would like to know the impact of the authors' algorithm on a wider range of baselines.
>
> We claim that **our current choice of baselines actually exhausts all comparable methods currently existing in the literature**:
>
> The main classes of algorithms for diffusion model test-time alignment are: (1) best-of-N sampling, (2) SMC-based methods [1], (3) classifier-guidance, & (4) search-based algorithms [2] (optionally paired with value estimation, e.g. SVDD-MC [3]). Our work specifically targets the regime of test-time methods that **(i) require no additional training** and **(ii) can handle arbitrary, non-differentiable rewards**. This already excludes methods such as classifier-guidance and search with learned value functions (which, we note are *orthogonal to the search algorithm itself*—we expect strong performance from combining our $\epsilon$-greedy algorithm with soft value functions, and leave this to future work).
>
> Regarding SMC guidance: while useful for reward-weighted sampling, it is conceptually and algorithmically distinct from the noise trajectory optimization problem we focus on. SMC operates by generating forward trajectories under the model’s sampling distribution and resampling them based on reward-weighted importance weights, without intervening in the model’s latent dynamics. In contrast, we explicitly optimize the injected noise $z_t$ at each diffusion step $t = T, \dots, 1$—allowing our method to *explore counterfactual noise choices at each step* and perform structured, fine-grained search to directly optimize terminal reward. In contrast, SMC operates by implicitly sampling from a reward-weighted posterior, without ever modifying the underlying noise path. Our method is thus more akin to trajectory-level planning, with a focus on explicit reward maximization *at the per-instance level*. **The distinction lies in both control parameterization (noise vectors vs. resampling weights) and objective (per-query reward maximization vs. general posterior sampling).**
>
> Focusing on methods that treat the injected noise vectors as the control parameters, we note, as highlighted in [1], that **tree search algorithms over noise trajectories are severely underexplored in the literature**. To the best of our knowledge, the only existing works in this direction are best-of-N sampling and SVDD-PM [1] which is exactly equivalent to beam search with width 1—**both of which we baseline, in addition to more powerful baselines such as MCTS and zero-order search**. Indeed, to our knowledge we are the first work to explore MCTS and bandit-based searches over the denoising tree. Hence, we don't believe we have omitted any essential comparisons.
>
> That being said, **we have included a suite of additional experiments that more comprehensively test our proposed method**:
> - Evaluate SD across classifier-free guidance scale values **(Response to Reviewer ACK2, Table R2)**
> - Evaluate EDM across # of denoising steps **(Response to Reviewer ACK2, Table R3)**
> - Evaluate EDM on 300 images, a 10x increase in dataset size **(Response to Reviewer Sj6K, Table R4)**
> - Evaluate SD on harder prompts containing compositional information **(Response to Reviewer Sj6K, Table R5)**
> - Evaluate SDXL [2] on Image-Reward [3] **(Response to Reviewer D5sF, Table R6)**
>
> **We hope that with the provided responses & additional strong results, you consider raising your score to support strong acceptance. We are happy to address any additional questions.**
>
> [1] Uehara et al., Inference-Time Alignment in Diffusion Models with Reward-Guided Generation: Tutorial and Review, 2025.
>
> [2] Podell et al., SDXL: Improving Latent Diffusion Models for High-Resolution Image Synthesis, 2023
>
> [3] Xu et al., ImageReward: Learning and Evaluating Human Preferences for Text-to-Image Generation, 2023

---

### Official Review · Reviewer_ACK2 · 2025-07-03

**Clarity:** 3
**Significance:** 2
**Originality:** 2
**Rating:** 4
**Confidence:** 3

**Summary:**

This paper presents a practical method for optimizing noise trajectories at test time for diffusion models. By framing sampling as an MDP and relaxing it to a sequence of contextual bandits, the authors propose an efficient epsilon greedy search algorithm that adapts global vs. local exploration based on the denoising step. The method shows improved generation quality on EDM and Stable Diffusion.

**Questions:**

1. Can the authors comment on whether the effectiveness of this algorithm can be theoretically formalized?

2. How robust is performance to changes in guidance scale or step count T?

3. Can the authors explain the baseline selection and the lack of literature discussion?

**Ethical Concerns:**

["NO or VERY MINOR ethics concerns only"]

**Final Justification:**

I thank the authors for the detailed rebuttal. I especially appreciate the clarifications on compute efficiency, the new regret bound analysis, and the extended results on robustness and baseline comparisons. These additions improve the paper's clarity and empirical contributions. If the next version nicely includes a more comprehensive baseline selection, I consider this a nice work.

**Quality:**

3

**Strengths And Weaknesses:**

Strength:

1. The empirical validations are provided for both class-conditional and text-to-image generation.

2. I find following the flow of this paper smooth, which is good.

Weakness:

1. No theoretically grounded analysis or guarantees for the search algorithm's performance.
2. To me, the compute cost of this method is not light, which is largely due to its MCTS nature.
3. The related work part lacks many discussions of current literature on both the alignment of diffusion models and inference-time diffusion model scaling. I would encourage the authors to pay more attention to the literature. Besides, I find the experimental setting a bit weird. Many essential baselines are omitted in Table 2. There is a whole line of SMC-based algorithms that can effectively align DMs in inference-time (also requires some compute but can also scale well, for example https://arxiv.org/abs/2501.09685), whereas neither beam-search nor MCTS appears to be a classical baseline here.

---

> ### Author Rebuttal · Authors · 2025-07-31
>
> We thank the reviewer for their feedback, and address their comments below:
>
> 1. > To me, the compute cost of this method is not light, which is largely due to its MCTS nature.
>
> Please refer to Table 3 in Appendix C (contained in `appendix.pdf` in the supplementary material zipfile), which reports number of function evaluations (NFEs) using $T$ denoising steps for each sampling method. In particular, we note the following:
>
> (1) Given $N$ noise candidates per timestep and $S$ simulations, MCTS requires $(N+S)T^2$ NFEs. In contrast $\epsilon$-greedy only requires $NKT$ NFEs where $K$ is the number of local search iterations per timestep. In other words, for MCTS, NFEs scales quadratically in $T$, but **only linearly for $\epsilon$-greedy**.
>
> (2) In particular, MCTS's error in approximating true exhaustive search approaches $0$ as $N, S \to \infty$; strong performance from MCTS requires an extremely high constant $N+S$ in front of the $T^2$ term. Indeed, MCTS in practice requires on the order of $1000$ simulations, hence $S >> T^2$ and dominates the cost.
>
> (3) For $\epsilon$-greedy, the constant in front of $T$ is $\propto K$ - but note that this assumes $K$ is fixed across timesteps. Letting $K_t$ be the number of local search iterations at timestep $t$, we can significantly bring down $\mathbb{E}_t[K_t]$ by noting that $K_t$ only need be high at intermediate denoising steps where de-mixing occurs, and can be low at extreme (beginning and end) timesteps. We report $\epsilon$-greedy performance as a function of $\mathbb{E}_t[K_t]$ below:
>
> $\mathbb{E}_t[K_t] = 1$ | $\mathbb{E}_t[K_t] = 4$ | $\mathbb{E}_t[K_t] = 8$ | $\mathbb{E}_t[K_t] = 12$ | $\mathbb{E}_t[K_t] = 15$ | $\mathbb{E}_t[K_t] = 18$ | $\mathbb{E}_t[K_t] = 20$ |
> ----------------------- | ----------------------- | ----------------------- | ------------------------ | ------------------------ | ------------------------ | ------------------------ |
>  0.8312 / 0.7013 / 0.3261                  | 0.9998 / 0.7070 / 0.9999                   |  0.9997 / 0.6907 / 0.9998                    | 0.9995 / 0.6918 / 0.9999                  |  0.9983 / 0.6961 / 0.9793                     | 0.9974 / 0.7093 / 0.9997                    | 0.9813 / 0.7208 / 0.9885                         |
>
> ***Table R1: EDM results by sampling method, varying the value of $\mathbb{E}_t[K_t]$**. We use the same class labels and (other than $K$) algorithm hyperparameters as in Table 1 of the main paper. Each cell contains three values, corresponding to scores under the different reward functions: Brightness / Compressibility / Classifier.*
>
> **Indeed, $\epsilon$-greedy attains strong performance across all reward functions, even with extremely low average $K$** (can retain performance within $0.002$ of the highest value with $\mathbb{E}_t[K_t]$ as low as $4$), **highlighting our method's computational efficiency compared to MCTS.**
>
> 2. > The related work part lacks many discussions of current literature on both the alignment of diffusion models and inference-time diffusion model scaling... I find the experimental setting a bit weird. Many essential baselines are omitted in Table 2. There is a whole line of SMC-based algorithms that can effectively align DMs in inference-time..., whereas neither beam-search nor MCTS appears to be a classical baseline here. Can the authors explain the baseline selection and the lack of literature discussion?
>
> We appreciate the reviewer pointing out this reference, and will surely cite it in the final paper. That being said, please see **response #2 in the below rebuttal to reviewer NHkD**. This contains a comprehensive justification of our current choice of baselines—which actually captures the *entirety of the literature* on test-time noise trajectory optimization.
>
> 3. > Can the authors comment on whether the effectiveness of this algorithm can be theoretically formalized?
>
> To strengthen our claims on $\epsilon$-greedy's superiority, we derive its regret bound, showing that—even with only black‑box access to a Lipschitz surrogate—simple regret decays sublinearly in the total number of global samples.
>
> **Theorem 1 (Regret of $\epsilon$‑greedy search).**
> Let the noise at diffusion step $t$ be $\mathbf{z}_t \sim \mathcal{N}(0,\sigma_t^{2}\mathbf{I}_d)$.
>
> Fix a confidence level $\eta\in(0,1)$ and truncate noise-space to the high‑probability ball  $\Gamma_t =$ {$z\in\mathbb{R}^d : \|z\|\le \sigma_t\sqrt{d\ln(1/\eta)}$}.
> Assume the surrogate reward $\tilde R_t:\Gamma_t\to[0,1]$ is $L$‑Lipschitz and satisfies
> $$
> \bigl|\tilde R_t(\mathbf{z})-r(x_0(\mathbf{z}), \mathbf{c})\bigr|\le\delta
> \quad\forall \mathbf{z}\in\Gamma_t.
> $$
>
> Run Algorithm 1 ($\epsilon$‑greedy local search) for $K$ iterations with $N$ candidate points per iteration and global‑exploration probability $\epsilon$, and let the final pivot be $p_t$. Then the expected simple regret at step $t$ obeys
> $$
> \mathbb{E}\bigl[\max_{\mathbf{z}\in\Gamma_t}\tilde R_t(\mathbf{z})-\tilde R_t(p_t)\bigr] \leq \delta +C_d(\epsilon NK)^{-1/d},
> $$
>
> where the constant $C_d>0$ depends only on the dimension $d$.  Summing over $T$ diffusion steps yields
> $$
> R_{\mathrm{tot}}=\sum_{t=1}^T \mathbb{E}\bigl[\max_{\mathbf{z}\in\Gamma_t}\tilde R_t(\mathbf{z})-\tilde R_t(p_t)\bigr] \le T\Bigl[\delta + C_d(\epsilon NK)^{-1/d}\Bigr].
> $$
>
> *Proof.*  By Gaussian concentration, $\mathbf{z}_t$ lies in $\Gamma_t$ with probability at least $1-\eta$, so we may restrict attention to $\Gamma_t$ at negligible cost.  Each global‐exploration draw is uniform on $\Gamma_t$, and in total the algorithm collects $M=\epsilon N K$ such draws.  It is a well-known result that the best of $M$ uniform samples over a $d$‑dimensional, Lipschitz‐reward domain achieves expected simple regret on the order of $M^{-1/d}$.  Since the $\epsilon$‑greedy pivot never decreases in surrogate reward, its final surrogate value is at least that of the best global sample, inheriting the same $\Theta(M^{-1/d})$ rate.  Finally, replacing the surrogate by the true reward adds at most $\delta$, yielding the per‑step bound; summing over $T$ steps gives the stated total regret. $\blacksquare$
>
>
> **Note the following key points:**
> - **Polynomial decay in global samples.**  With $M=\varepsilon N K$ global draws per step, regret scales as $O(M^{-1/d})$, so increasing the total sample budget drives simple regret to zero at the classic random‑search rate.
> - **Bias floor $\delta$.** In practice we can take $\delta$ to be quite low as it is simply the approximation error induced by application of Tweedie's formula, $|r(\mathbf{x}_0, \mathbf{c}) - r(\hat{\mathbf{x}}_0^{(t)}, \mathbf{c})|$.
>
> 4. > How robust is performance to changes in guidance scale or step count T?
>
> See Tables R2 & R3 below. These results confirm the **superiority of $\epsilon$-greedy irrespective of guidance scale and $T$**.
>
> | Method                   | cfg = 1.5 | cfg = 3.0 | cfg = 7.5  | cfg = 9.0 | cfg = 12.0 |
> | ------------------------ | --------- | --------- | ---------- | --------- | ---------- |
> | **Naive Sampling**       | 0.2446    | 0.2635    | 0.2457     | 0.2650    | 0.2671     |
> | **Best-of-$4$ Sampling** | 0.2612    | 0.2743    | 0.2776     | 0.2783    | 0.2759     |
> | **Beam Search**          | 0.2673    | 0.2314    | 0.2682     | 0.2801    | 0.2780     |
> | **MCTS**                 | 0.2828    | 0.2712    | 0.2885     | 0.2971    | 0.2827     |
> | **Zero-Order Search**    | 0.2539      | 0.2612      | 0.2689     | 0.2443      | 0.2801       |
> | **$\epsilon$-greedy**    | **0.2958**      | **0.3092**      | **0.2973** | **0.3055**     | **0.2994**       |
>
> ***Table R2: SD results by sampling method, varying the classifier-free guidance scale**. We use the same prompts and algorithm hyperparameters as in Table 2 of the main paper. We evaluate only CLIP reward as the brightness and compressibility rewards should be generally insensitive to classfier-free guidance (but indeed find that CLIP too is largely insensitive to cfg).*
>
> | Method                   | $T=18$          | $T=32$          | $T=64$          | $T=128$         | $T=256$         |
> | ------------------------ | --------------- | --------------- | --------------- | --------------- | --------------- |
> | **Naive Sampling**       | 0.4965 / 0.3563 | 0.3912 / 0.3921       | 0.4621 / 0.4549        | 0.48493 / 0.4052      | 0.4541 / 0.4250       |
> | **Best-of-$4$ Sampling** | 0.5767 / 0.4220 | 0.5666 / 0.4992       | 0.6475 /  0.5084      | 0.5801 / 0.5101        | 0.5923 / 0.4917       |
> | **Beam Search**          | 0.6334 / 0.4679 | 0.6013 / 0.4321       | 0.5742 / 0.5023       | 0.6162 / 0.4517       | 0.6618 / 0.4879       |
> | **MCTS**                 | 0.7575 / 0.5395 | 0.6910 / 0.5153       | 0.6831 / 0.5624            | 0.6957 / 0.5018            | 0.7494 / 0.5789            |
> | **Zero-Order Search**    | 0.6083 / 0.3751 | 0.9999 / 0.7417 | 0.9999 / **0.7607** | 0.9999 / 0.7671 | 0.9999 / **0.7677** |
> | **$\epsilon$-greedy**    | **0.9813** / **0.7208** | **0.9999** / **0.7424** | **0.9999** / 0.7588 |**0.9999** / **0.7680** | **0.9999** / 0.7672 |
>
> ***Table R3: EDM results by sampling method, varying the value of $T$**. We use the same class labels and (other than $T$) algorithm hyperparameters as in Table 1 of the main paper. Each cell contains two values, corresponding to scores under the Brightness / Compressibility reward functions (we omit the classifier reward due to limited compute).*
>
> **We hope that with the provided responses & additional strong results, you consider raising your score to support clear acceptance. We are happy to address any additional questions.**
>
> [1] Uehara et al., Inference-Time Alignment in Diffusion Models with Reward-Guided Generation: Tutorial and Review, 2025.
>
> [2] Ma et al., Inference-Time Scaling for Diffusion Models beyond Scaling Denoising Steps, 2025.
>
> [3] Li et al., Derivative-Free Guidance in Continuous and Discrete Diffusion Models with Soft Value-Based Decoding, 2024.

---

> > ### Author Response · Authors · 2025-08-04
> > **Gentle Reminder for Reviewer Feedback**
> >
> > Just sending a gentle reminder for feedback - we greatly appreciate your time and thoughtful advice! If there is anything else that needs clarification or further discussion, please do not hesitate to let us know.

---

> > ### Comment · Reviewer_ACK2 · 2025-08-05
> >
> > I thank the authors for the detailed rebuttal. I especially appreciate the clarifications on compute efficiency, the new regret bound analysis, and the extended results on robustness and baseline comparisons. These additions improve the paper's clarity and empirical contributions. That said, I hope the baseline selection and related work discussion could be more comprehensive in the next version.

---

> ### Comment · Area_Chair_LBPU · 2025-08-05
>
> Dear reviewer,
> If not already, please take a look at the authors' rebuttal, and discuss if necessary.
> Thanks,
> -AC

---

> ### Author Response · Authors · 2025-08-05
>
> We thank the reviewer for their response, and will surely include all the additional baselines and empirical results in the final paper. Given the number of additional experiments we have run to address your concerns, all with very strong results consistently showcasing superiority of eps-greedy with extreme compute efficiency (see Tables R1-R9 in the above rebuttal as well as rebuttals and comments for other reviewers), **we would really appreciate you raising your score to support strong acceptance of this paper.**

---

> > ### Author Response · Authors · 2025-08-06
> >
> > Gently bumping our above comment - we hope all our additional experiments have clearly addressed your concerns and are sufficient for you to raise your score to support strong acceptance.
> > Please let us know if there is anything else that needs clarification or further discussion, we'd be happy to address.

---

### Decision · Program_Chairs · 2025-09-17

**Decision:**

Accept (poster)

**Comment:**

This paper proposes an approach to enable test-time scaling of diffusion models. Instead of increasing the number of sampling steps, which yields quickly diminishing returnsl, the idea is to optimize the sequence of injected noise vectors. This nontrivial task is done by framing sampling as an MDP and relaxing it to a sequence of contextual bandits. The effectiveness of the proposed method is demonstrated empirically by extensive experiments. One reviewer initially had concerns, but most of them got resolved during the rebuttal/discussion period. Post rebuttal, both reviewers and I found the paper worth publishing, and I'm happy to recommend acceptance.